

**Opinion: Papers that shaped Tropospheric Chemistry**
Paul S. Monks,[1] A.R. Ravishankara,[2] Erika von Schneidemesser[3] and Roberto Sommariva[1]
1. School of Chemistry, University of Leicester, University Rd., Leicester, LE1 7RH, UK.
2. Departments of Chemistry and Atmospheric Science, Colorado State University, Fort Collins,
9       Colorado, USA.
3. Institute for Advanced Sustainability Studies, Berlinerstrasse 130, 14467 Potsdam, Germany.
**Abstract**
Which published papers have transformed our understanding of the chemical processes in the
troposphere, and shaped the field of atmospheric chemistry? By way of expert solicitation and
interactive peer-review, this paper explores the influence of the ideas in peer-reviewed articles based
on the input from our community of atmospheric scientists. We explore how these papers have
shaped the development of the field of atmospheric chemistry, and identify the major landmarks in
the field of atmospheric chemistry through the lens of those papers' impact on science, legislation and
environmental events. We also explore the ways in which one can identify the papers that have most
impacted the field and discuss the advantages and disadvantages of the various approaches. Our work
highlights the difficulty of creating a simple list and we explore the reasons for this difficulty. The
paper also provides a history of the development of our understanding of tropospheric chemistry and
points some ways for the future.

## 1. Introduction

Air quality and anthropogenic climate change are two environmental issues of current importance to society. Atmospheric composition is central to both these issues. The atmosphere, and its components, supports life on Earth. In turn, the atmosphere is affected by human population growth and industrialization, as well as all the consequences of those changes. The changes in atmospheric composition also influence the ecosystem on which humans rely.

Air pollution (née composition) and its impacts have a history stretching back to antiquity – see for example the expositions in (Brimblecombe, 1987;Fuller, 2018;Jacobson, 2002;Stern, 1968;Sportisse, 2010;Preining and Davis, 1999;Fowler et al., 2020) and others. Changes in atmospheric composition, with negative impacts particularly on human health (Lelieveld et al., 2015;Landrigan et al., 2018), ecosystems (Fowler et al., 2009) and latterly climate (see for example, (Fiore et al., 2012;von Schneidemesser et al., 2015)), have become primary global concerns during the latter part of the 20[th] and the 21[st] centuries. As an academic subject, air pollution has mostly been systematically studied only since the mid-late 20[th] Century. There have been several recent reviews, (e.g., (Brasseur et al., 2003;Monks et al., 2009;Ravishankara et al., 2015;Ravishankara, 2003)), which have mapped the growth of atmospheric chemistry, but it is not only peer-reviewed papers that provide relevant overviews. It is important to note that when dealing with the development of this subject (or any scientific subject for that matter), much of the baseline knowledge is embodied in textbooks, which for many are the entry point to and the primary reference for the topic (e.g. (Jacob, 1999;Wayne, 2000;Finlayson-Pitts and Pitts, 2000;Seinfeld and Pandis, 2006;Brasseur et al., 1999)).

Figure 1 shows the number of peer-reviewed papers by year that mentioned the phrase "atmospheric chemistry" in the text, as catalogued by the Scopus bibliographic database (https://www.scopus.com/). It shows a growth in the later 1970s from around a hundred papers a year to approximately 4,000 a year currently, with a large increase especially over the past two decades. Of course, many more papers discuss atmospheric chemistry, or are relevant to it, without explicitly mentioning these words!

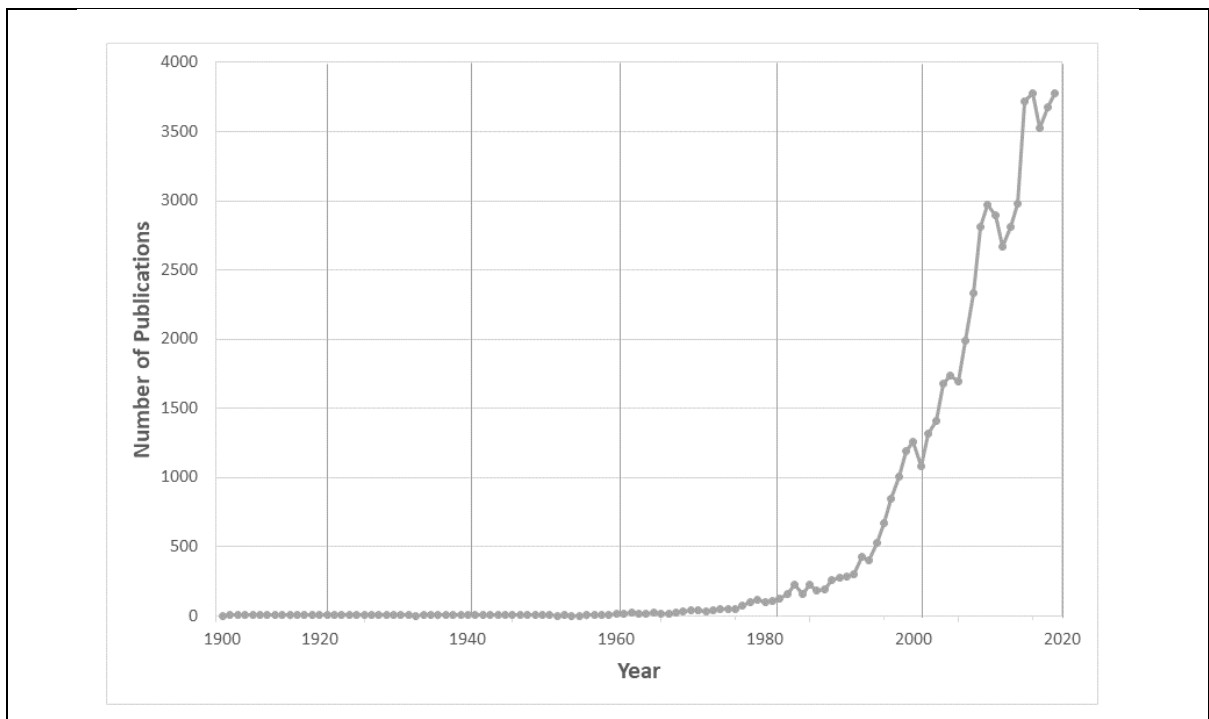

Figure 1 – Number of publications per year in a SCOPUS search on the phrase "atmospheric chemistry" (compiled in June 2020).

This paper aims to recognize and highlight some of the most influential peer-reviewed articles that
have shaped this field. There were many pivotal scientific discoveries and there were many papers
that spurred action and further research. What were the significant discoveries that shaped the
atmospheric chemistry of today? And how do we narrow down the list contributions to highlight the
most impactful ones?
There are many ways to choose the papers that described discoveries and influenced atmospheric
composition and chemistry. Here we have assembled a compilation of influential articles. Our goal is
not to show what makes a 'great' paper, which depends not only on the science, but also the quality
of the writing, readability, structure of the written work, and the reader – all criteria that are highly
subjective. Nor are we aiming only for those papers that led to policy and management actions.
Instead, we try to reflect on *the papers' science and content* and the influence of the ideas in those
papers on our community of scientists and on the field's development. Our approach is to present our
thoughts – informed by the solicitation for input from colleagues in the field – and share what we
think are the fundamental discoveries and developments, start a discussion, and allow others to build
on, reinforce or critique our work.
In addition to peer-reviewed papers, and the above mentioned textbooks,  we have of course other
mediums through which we communicate (have communicated) with our peers. These  include
scientific reports, conferences, and meetings. In addition, we have scientific assessments and
evaluations, which often get more scrutiny and review than the peer-reviewed papers they include.
These days, other communication media, such as social media, have also become prevalent as formats
for exchange both within the scientific community and with a broader audience. However, the entire
community cannot attend all conferences and meetings, the scientific reports are not always
accessible and often not peer-reviewed, and the assessments are often driven more by policy needs
rather than by scientific discoveries. Publishing peer-reviewed papers is the closest we come to
reaching the whole community. We do note that, despite its known issues, the peer-reviewed
literature is still considered the gold standard for quality and reliability. For these reasons, we discuss
only peer-reviewed papers here, although we aim to communicate the overarching scientific advances
that shaped the field.

### 1.1. How were the papers selected?

Easily measurable criteria, like the number of citations, are one metric. However, this approach
favours papers of a particular vintage and not necessarily the earlier or later papers.  Besides, there
are several drawbacks to these simple and objective criteria. Citations tend to go down when
something is assumed to be common knowledge and makes it into textbooks or compilations. For
example, nobody cites Priestley for discovering oxygen or Schönbein for discovering ozone, whenever
atmospheric composition is mentioned. Indeed, some of the central concepts of atmospheric
chemistry and physics are considered to be common knowledge, and their origins are taken for
granted. The number of citations will also be influenced by the journal in which a paper is published,
and quite often (we hate to say this) also depends on who else cited them and in which journal they
were cited. Citations also depend on how many people are otherwise researching a particular subject.
Furthermore, critical assessments and expert data evaluations suppress the citation of the original
papers. This is particularly the case, for example, for papers on chemical kinetics and photochemistry,
whereby people tend to simply cite the data evaluations such as National Aeronautics and Space
Administration Jet Propulsion Laboratory (NASA/JPL, https://jpldataeval.jpl.nasa.gov/)  or
International Union of Pure and Applied Chemistry (IUPAC, http://iupac.pole-ether.fr/) panel reports.
Similarly, people often cite the quadrennial ozone layer depletion and Intergovernmental Panel on
Climate Change (IPCC) assessments, thereby obfuscating the underlying original papers. Other types
of papers, such as reviews, tend to get an excessive number of citations (for understandable reasons).
Lastly, we cannot overlook the influence of journal availability in different parts of the world. This
availability is exacerbated when journal costs go up, and not everybody can access new papers.
Nevertheless, there is still a relevance to the number of citations of a paper.  We show, for example,
the 10 most-cited papers when we were to search on the combination of words "atmospheric and
chemistry" in Table 1.
For all the above reasons, we decided to use a different approach here. We solicited the scientific
community to obtain input from the experts in the field. To accomplish this, we put out a call through
the International Global Atmospheric Chemistry (IGAC) (Melamed et al., 2015) project to its contacts
and thereby engaged a broad audience. Despite the broad audience of IGAC, the vast majority of
responses came from scientists in North America and Europe. An initial list of influential papers was
established by combining the replies received from the expert solicitation to evaluate the most
nominated papers. In addition, a variety of perspectives were assembled for the writing team,
including different career stages, nationalities, and genders. Despite all these efforts, the selection
methods will still inevitably create bias that cannot be escaped. Therefore, in many respects, the
chosen papers are not supposed to be a definitive list, but rather a compilation that allows researchers
to discuss and reflect on what makes impactful science, and maybe ponder on the landmarks in our
subject. Furthermore, we hope that the end product can provide an interesting history and context to
those who are joining the community and document the current "perception" of what are the most
important papers.
We have noted the drawbacks in our methodology simply to present upfront some of the limitations
of what we did in this paper. However, we hope that others will find this work relevant and engaging.
Through an open and active peer-review process, we were able to obtain the perspective of a broader

community more reflective of the global composition of the field. To facilitate this, the paper was published first as an open-access, discussion paper that included a public comment period. We hope that this approach overcame some of the limitations and reservations we expressed earlier. We thank all reviewers for their contributions and help in determining the final shape of this overview.

*1.2. Scope of work*

As with the selection method, one can debate the scope and the methodology for a work such as this. Still, the boundaries we have drawn encompass studies that have shaped our understanding of the atmosphere and the underlying chemical and physical processes, focusing mostly on the troposphere. This includes modelling, field measurements, remote sensing, and laboratory studies (Abbatt et al., 2014). We have also included atmospheric interactions with the biosphere, cryosphere, and hydrosphere.

We selected 2010 as the cut-off year. Our rationale is that for a paper to have been influential in the whole field it must be at least ten years old and thus had time to accrue recognition. We recognize that important papers in newer areas of endeavour are disadvantaged by this criterion. Examples include the subjects of SOA formation (Ehn et al., 2014;Crounse et al., 2013), the chemistry of Criegee intermediates (Welz et al., 2012;Mauldin Iii et al., 2012), galactic rays induced aerosol particle nucleation (Kirkby et al., 2011), and air pollution-climate connections (Shindell et al., 2012). Influential assessments such as bounding black-carbon (Bond et al., 2013) are also missed. However, these areas will undoubtedly be recognized in the coming decades. The ten-year window also allows the scientific community to have had extensive input on a paper's validity, i.e., meeting the criterion of "standing the test of time."

The papers have been grouped into the following general categories and are presented as such in Section 2.

1. *Foundations*
2. *Aerosols and Clouds*
3. *Secondary Organic Aerosols*
4. *Chemical Kinetics, Laboratory Data and Chemical Mechanisms*
5. *Heterogeneous and Multiphase Chemistry*
6. *Chemical Models*
7. *Tropospheric Ozone*
8. *Nitrogen Chemistry*
9. *HOx Chemistry*
10. *Nightime Chemistry*
11. *Halogen Chemistry*
12. *Volatile Organic Compounds*
13. *Biogenic Emissions and Chemistry*
14. *Biomass Burning*
15. *Emissions and Deposition*
16. *Chemical Transport*
17. *Satellites and the Troposphere*
18. *Stratospheric Chemistry*
19. *Other issues that influenced tropospheric chemistry*

The groups were chosen to reflect the main areas of research or endeavour, recognizing that this
division could be done in several different ways. There is no assumed equivalence in these groups
regarding their perceived or real importance or impacts. In the following, we discuss the papers in
each group to show why they have been nominated and to put them in the historical context of the
development of atmospheric chemistry as a discipline.

## 2. Survey of Areas

### 2.1. Foundations

Atmospheric chemistry has some long-standing and deep roots. However, it blossomed in the second half of the 20th century following concerns about ozone layer depletion and various forms of tropospheric pollution, such as the Los Angeles smog, London smog, and acid precipitation (Table 2). Many note John Dalton's early contributions on the proportion of gases in the atmosphere (Dalton, 1805) and John Tyndall's Bakerian lecture on radiation and gases (Tyndall, 1861) as among the first studies in this field. The work of Arrhenius "On the Influence of Carbonic Acid in the Air upon the Temperature of the Ground" (Arrhenius, 1896) and the subsequent paper of Callendar, "The artificial production of carbon dioxide and its influence on temperature" (Callendar, 1938), laid the groundwork for the linkage between atmospheric chemistry and climate. Concerning aerosols, the seminal work of John Aitken (Aitken, 1888) "On the number of dust particles in the atmosphere" details early work to count the number per cubic centimeter in various indoor and outdoor environments. It is interesting to note that physiologists looking at the number of live germs in the air stimulated Aitken's work. The later work of Köhler (Köhler, 1936) which explored cloud droplet nucleation remains the basis for later work (see the *Aerosols and Clouds* section). The start of atmospheric chemistry as a distinct discipline probably arrived with Chapman's chemical theory of the stratospheric ozone layer in 1930 (Chapman, 1930), which will be further discussed in the *Stratospheric Chemistry* section. This study heralded the importance of atmospheric chemistry on a global scale.

In analyzing the influential papers on atmospheric composition, one cannot help but note the relationship between these papers and the most significant contemporary environmental issues (Table 2). The first of these was the Los Angeles smog, which had its European counterpart, the London "Pea-Soup" (Brimblecombe, 1987). The two events, which in chemical terms have no equivalence, had comparable impacts on public health and opinion. The oft recognized work of Haagen-Smit (Haagen-Smit and Fox, 1954;Haagen-Smit, 1952;Haagen-Smit et al., 1953) in the early 1950s on the Los Angeles smog was the first to coin the term "air pollution" in the modern era. Haagen-Smit showed that automobile exhaust gases can form ozone in the air and should, therefore, be considered a definite source of smog. Figure 2, redrawn from Haagen-Smit (1952), shows a schematic presentation of the reactions in polluted air leading to smog. Notably, the basic features of tropospheric chemical processes, as we understand them today, were already recognized in these early papers, and they showed how ozone could be chemically produced in the troposphere. Brasseur has documented these findings in a very thorough review (Brasseur et al., 2003).

It is widely recognized that both Crutzen (Crutzen, 1973a;Crutzen, 1973b) and Chameides and Walker (1973) found that similar "smog reactions" oxidize methane ($CH_4$) and carbon monoxide (CO) to produce substantial amounts of ozone in remote regions of the atmosphere. They estimated chemically produced ozone to be much greater than that transported from the stratosphere, which was believed to be the primary source of this chemical in the troposphere at that point. A few years earlier, in 1970, Hiram Levy II suggested that the hydroxyl radical, which provides the dominant oxidation mechanism in the troposphere, was formed in unpolluted air by the same mechanism that had been described as occurring in polluted air (Levy, 1971). This paper by Levy (1971) is recognized by many as the first description of the chemistry of the lower atmosphere involving hydroxyl radical

reactions of methane and carbon monoxide, hydroperoxyl radicals, and the photolysis of ozone and
formaldehyde as radical sources. In particular, he recognized that the very short-lived electronically
excited oxygen atom ($O^1D$) is a possible source of the hydroxyl radical (OH), an idea now well
established.
Around the same time, Weinstock (1969) explained how cosmic rays lead to the production of
radiocarbon dioxide ($^{14}CO_2$), which is incorporated into living plants. This process requires a rapid
turnover of radiocarbon monoxide ($^{14}CO$), which was unexpected because the lower atmosphere was
thought to be a "chemical desert". Instead, carbon monoxide appeared to have a turnover time of
about one-tenth of a year, primarily driven by hydroxyl radical oxidation. To some, this paper kicked
off the research which led to our present understanding of the atmospheric chemistry of the lower
atmosphere.

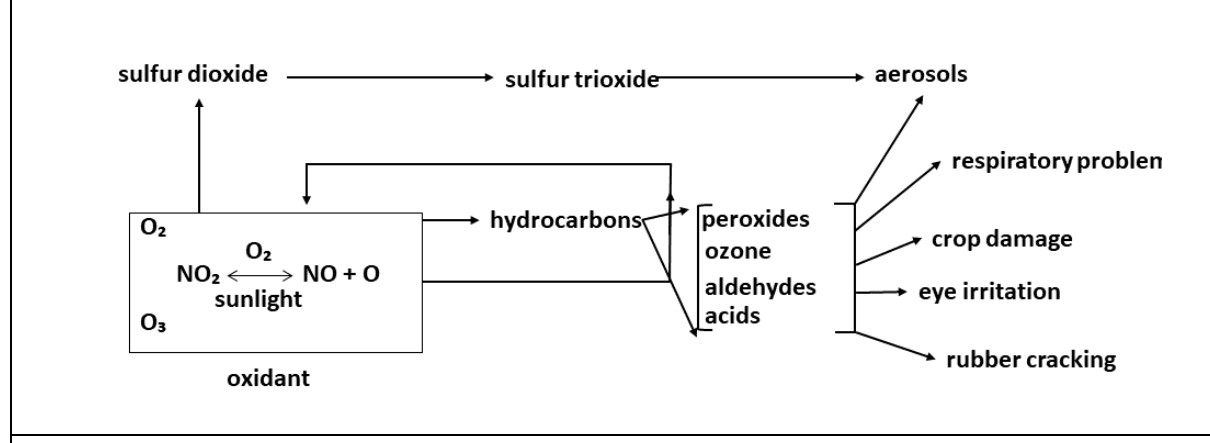

Figure 2 – Schematic representation of the reactions in polluted air leading to smog symptoms
(adapted from Haagen-Smit et al. (1953)).


It has been claimed that "acid rain was one of the most important environmental issues during the
last decades of the twentieth century" (Grennfelt et al., 2019) (see Table 2).  One of the reasons is that
acid rain first demonstrated that air quality was not merely a local issue but a regional issue and
showed that the atmosphere has no definite boundaries (Fowler et al., 2020). Although the case of
acid rain and its effects had been noted and reported by some earlier papers e.g., Odèn (1968), for
some, it is the paper by Likens and Bormann (1974) that made this issue known to the science
community at large. Other early papers (for example, from Urone and Schroeder (1969) and Penkett
et al. (1979)) also recognized the vital role of liquid-phase oxidation of sulphur dioxide ($SO_2$) by
oxidants such as hydrogen peroxide ($H_2O_2$) and ozone ($O_3$). Current estimates suggest that roughly
50% of the $SO_2$ oxidation in the lower troposphere occurs through liquid-phase reactions.
The story of lead in the atmosphere is a complex interplay between science, policy and economics
(Monks and Williams, 2020), where observations in snow (Murozumi et al., 1969) underpinned the
alarming growth and spread of lead pollution and latterly its demise (Boutron et al., 1991).
There is little doubt that one of the most impactful series of papers is that of the eponymous curve by
Keeling (Keeling, 1960;Keeling et al., 1979;Pales and Keeling, 1965), showing the steady rise in carbon
dioxide ($CO_2$) measured at Mauna Loa observatory (this work has continued uninterrupted by
NOAA/ESRL/CML over the past few decades). Keeling's work was built on the previously mentioned
work of Callendar (1938) who compared measurements of $CO_2$ at Kew, UK (1891-1901) with those in
the Eastern USA (1936-1938) and noted an increase in concentration. Although the gas in question is
$CO_2$, which is often seen only as climate gas, changes in its levels reflect the changing composition of
the atmosphere and the effects that it can have, and shows that the two subjects cannot be easily
separated. Furthermore, the increase in $CO_2$ is of central to ocean acidification, a topic not touched
upon here but nevertheless very important. The seminal paper by Ramanathan et al. (1985) that
highlighted the role of $CH_4$, chlorofluorocarbons (CFCs), and nitrous oxide ($N_2O$) for climate
strengthened the case for the inclusion of chemistry in the climate issue. In many respects, this close-
coupling between atmospheric chemistry and climate change was brought to the forefront with the
1995 Nobel Prize being awarded to Paul Crutzen, Mario Molina and F. Sherwood Rowland "for their
work in atmospheric chemistry, particularly concerning the formation and decomposition on ozone"[1]
(see *Stratospheric Chemistry* section) and, later, with the Nobel Prize to the IPCC.
The role of field campaigns, observations and the attendant models in shaping our understanding of
atmospheric chemistry should be recognised as foundational. In general, the adage that the
atmosphere is under-observed is still true.  Every time a new instrument has been developed to detect
a new chemical in the atmosphere, there have been significant advances (Heard, 2006).  One could
posit that the entire field of atmospheric science started because of detection and quantification of
oxygen and ozone in the Earth's atmosphere.  Some recent major advances in our field has been
through field measurement. For example, observation of the ozone depletion (including the ozone
hole), aerosol particles, free radicals, and stable molecules (including ozone layer depleting
substances, CO and methane) fundamentally changed the course of the field.  Organized systematic
probing of the atmosphere has been critical over the past four, or so, decades. Here again,
introduction of new instruments (optical, mass spectrometric, etc.) have been game-changing.  It is
also important to note and highlight the enormous contributions of satellite observations to provide
global coverage. Often, as field campaigns and their impact are spread across many papers it is difficult
to pull out their specific contributions.  Many of the early experiments, encompassing long-range
transport, biomass burning and aerosols, particularly using aircraft, have been detailed in (Melamed
et al., 2015). Assembling a large number of instruments on a large aircraft to simultaneously measure
an array of chemicals was pioneered by Davis (Davis, 1980) and has been a paradigm for field studies
ever since.
In addition to organized episodic field measurements, continual measurements of chemicals (often
called monitoring) has produced some of the most significant findings about the atmosphere.  For
example, continual monitoring of surface ozone from Paris or similar stations going back over a
hundred years or more has shown the trends in tropospheric pollution due to human activities (Volz
and Kley, 1988). The continual monitoring of the Antarctic ozone led to the discovery of the ozone
hole. The continual monitoring of $CO_2$ is the poster-child for climate change!  Much of this continual
monitoring has been carried out by national agencies and international partnerships. Examples include
the US National Oceanic and Atmospheric Administration (NOAA) Earth Systems Research Laboratory
(ESRL) Global Monitoring Laboratory's (GML) contributions (Montzka et al., 2007) and international
efforts, such as The Advanced Global Atmospheric Gases Experiment (AGAGE) network (Prinn et al.,
2001;Prinn et al., 1995), World Meteorological Organisation – Global Atmospheric Watch (WMO-

---

[1] https://www.nobelprize.org/prizes/chemistry/1995/press-release/

GAW) (WMO, 2017) and the Network for the Detection of Atmospheric Composition Change (NDACC)
(De Mazière et al., 2018).

293        *2.2. Aerosols and Clouds*

Aerosols in the atmosphere greatly influence both air quality and climate change; they are also a
significant media for composition change in the atmosphere. In this section we discuss three main
areas of research related to aerosols: (1) Understanding the mechanisms and atmospheric chemistry
processes that influence aerosol particle formation, nucleation, and growth, and how aerosols affect
composition; (2) The role of aerosols as cloud condensation nuclei and the influence that this process
has on climate; and (3) The impact of particulate matter on human health.  These areas are, however,
related and there is not always a clear division. *Secondary Organic Aerosols* (SOA) and *Heterogeneous*
*and Multiphase Chemistry* are discussed in the corresponding sections (2.3 and 2.5).
The roots of modern aerosol science lie, as previously discussed (see the 2.1. *Foundations* section), in
the works of Aitken (1888) and Köhler (1936) on the cloud droplet. Twenty years after Köhler's
research, Junge (1955) provided the power-law describing aerosol particle number and identified the
stratospheric aerosol layer, now dubbed the 'Junge layer'. Junge concluded: "A real step forward in
the understanding of the basic processes in air chemistry can be gained only if aerosol particles and
gases are measured simultaneously but separately, and if the aerosol particles, in turn, are separated
according to size." This suggestion has been a clarion call for atmospheric scientists ever since.
Junge and Ryan (1958) attempted to elucidate the formation of particles from gas-phase reactants,
particularly $SO_2$ and ammonia ($NH_3$), while Fitzgerald (1974) investigated the variation in aerosol
particle composition with particle size. They showed that cloud droplet size distribution was
insensitive to the specific soluble constituents. Twomey (Twomey, 1977, 1974) suggested that air
pollution gives rise to the whitening of clouds and influences the planet's radiative balance. He also
indicated that there is a connection between pollution aerosols and cloud reflectance (albedo). This
concept is now often referred to as the "Twomey effect." Twomey (1977) expanded on the 1974 work,
exploring the balance between the scattering *versus* absorption effect on incoming solar radiation. It
is on this basis that much of the current research on the role of aerosols *via* their direct and indirect
effects on climate has been built. Bolin and Charlson (1976) estimated that anthropogenic sulphate
aerosol from the US and Europe would lead to a global temperature decrease of 0.03-0.06 °C. They
recognized early on that "we are already approaching the time when the magnitude of the indirect
effects of increasing use of fossil fuel may be comparable to the natural changes of the climate over
decades and centuries."
In the early 1970s, Whitby and Knutson developed an instrument to measure particle size distribution
in the nanometer to micrometer range (Knutson and Whitby, 1975) – the well-known aerosol particle
mobility analyser. They used the measurements from this instrument to introduce a new formulation
of the formation and growth of atmospheric aerosol particle size modes – the "Whitby diagram" which
is now a common text book figure and is shown in Figure 3. The outcomes of this work show the
importance and influence of the development of new instruments that probe the atmosphere.

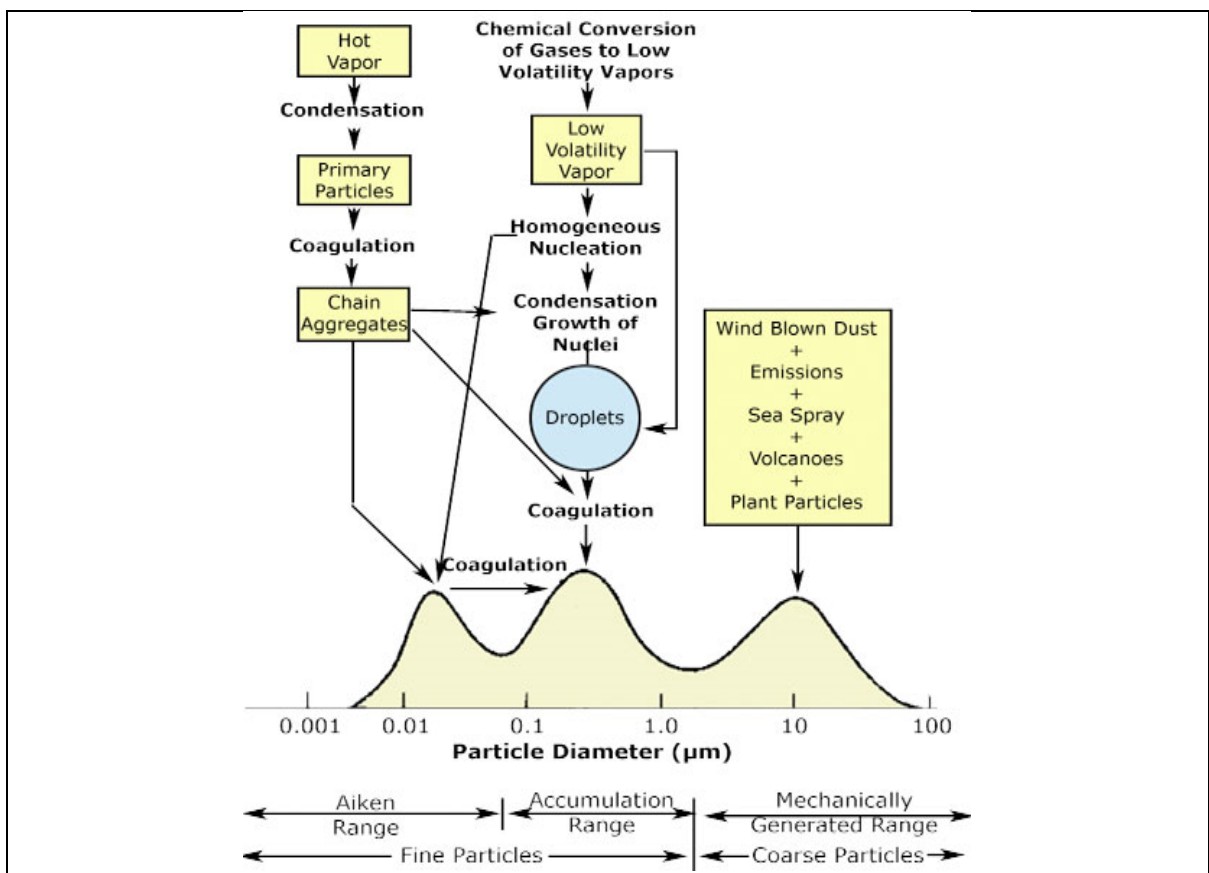

Figure 3 - Schematic of atmospheric aerosol particle size distribution showing the three modes, the main source of mass for each mode, and the principal processes involved in inserting mass to each mode along with the principal removal mechanisms (Whitby, 1978). (taken from https://serc.carleton.edu/NAGTWorkshops/metacognition/largeclasses.html, under CCC)


The CLAW hypothesis (the acronym taken from the surnames of the proposers Charlson, Lovelock,
Andreae, and Warren) (Charlson et al., 1987) further connected aerosol science to gas-phase
chemistry, specifically focused on the feedback loop between ocean ecosystems and Earth's climate.
This hypothesis built on earlier work by Lovelock et al. (1972) on the oxidation of marine
dimethylsulphide. Although the conclusions of Charlson et al. (1987) have been questioned (Quinn
and Bates, 2011), this paper highlighted the interconnections within atmospheric sciences, and
environmental sciences in general.
The work of Friedlander and co-workers (Stelson et al., 1979) further highlighted the role of liquid-
phase chemistry leading to aerosol particles. A key milestone in our understanding of sulfate
formation was the recognition that the reaction of the hydroxysulfonyl radical (HOSO$_2$) with oxygen
(O$_2$) is exothermic (Calvert et al., 1978) and leads to gas-phase sulphur trioxide (SO$_3$), contrary to what
was accepted at the time. Prior to this finding, there were major difficulties in understanding the
formation of gas-phase sulphuric acid (H$_2$SO$_4$) (Davis et al., 1979) from gas-phase SO$_2$ oxidation, an
essential step for the nucleation of new particles from the gas-phase in the atmosphere.
This area of research was further developed by Robbin and Damschen (1981) who investigated the
role of peroxide in the liquid-phase in oxidising SO$_2$ which was key to understanding the phenomenon
of acid rain. Graedel and Weschler (1981) reviewed the chemical transformations in atmospheric
aerosol particles and raindrops and extended the idea of Martin and Damschen. Stelson and Seinfeld
(1982) evaluated the thermodynamics of ammonium, nitrate, and sulphate aerosols, which was a
significant step in understanding particle formation and growth. Pankow's 1994 work (Pankow, 1994b,
a) on the absorption model of gas/particle partitioning of organic compounds in the atmosphere is of
fundamental importance for models to calculate the amounts of particulate matter (PM) formed and
their growth in the urban and regional air, and in the global atmosphere.
Charlson et al. (Charlson et al., 1990;Charlson et al., 1991) produced the first global estimate of the
direct aerosol effect that subsequently had a large impact on climate modelling. The role that aerosols
have on cloud condensation nuclei (CCN) and cloud albedo was also acknowledged, concluding that it
may be substantial. How substantial, however, was not quantified at that point because of a lack of
knowledge on the relationships involved. A few years later, Boucher and Lohmann (1995) provided an
estimate of the indirect effect of anthropogenic aerosols on climate. After many additional years of
study based on these foundations and analyses of radiative balance, the total radiative forcing by
anthropogenic aerosol is now estimated to be roughly -1.1 W/m$^2$ (IPCC, 2013), thereby solidifying the
importance of aerosols in climate change.
Building on the work of Whitby, Mäkelä et al. (1997) conducted continuous monitoring of particles at
a forest site in Finland. Beyond confirming the existence of three submicron particle size modes (the
nucleation, Aitken, and accumulation modes (see also Covert et al. (1996)), they also observed new
particle formation events. These events have been subsequently observed by others and are often
depicted in the literature using the famous "banana plots" (Figure 4).

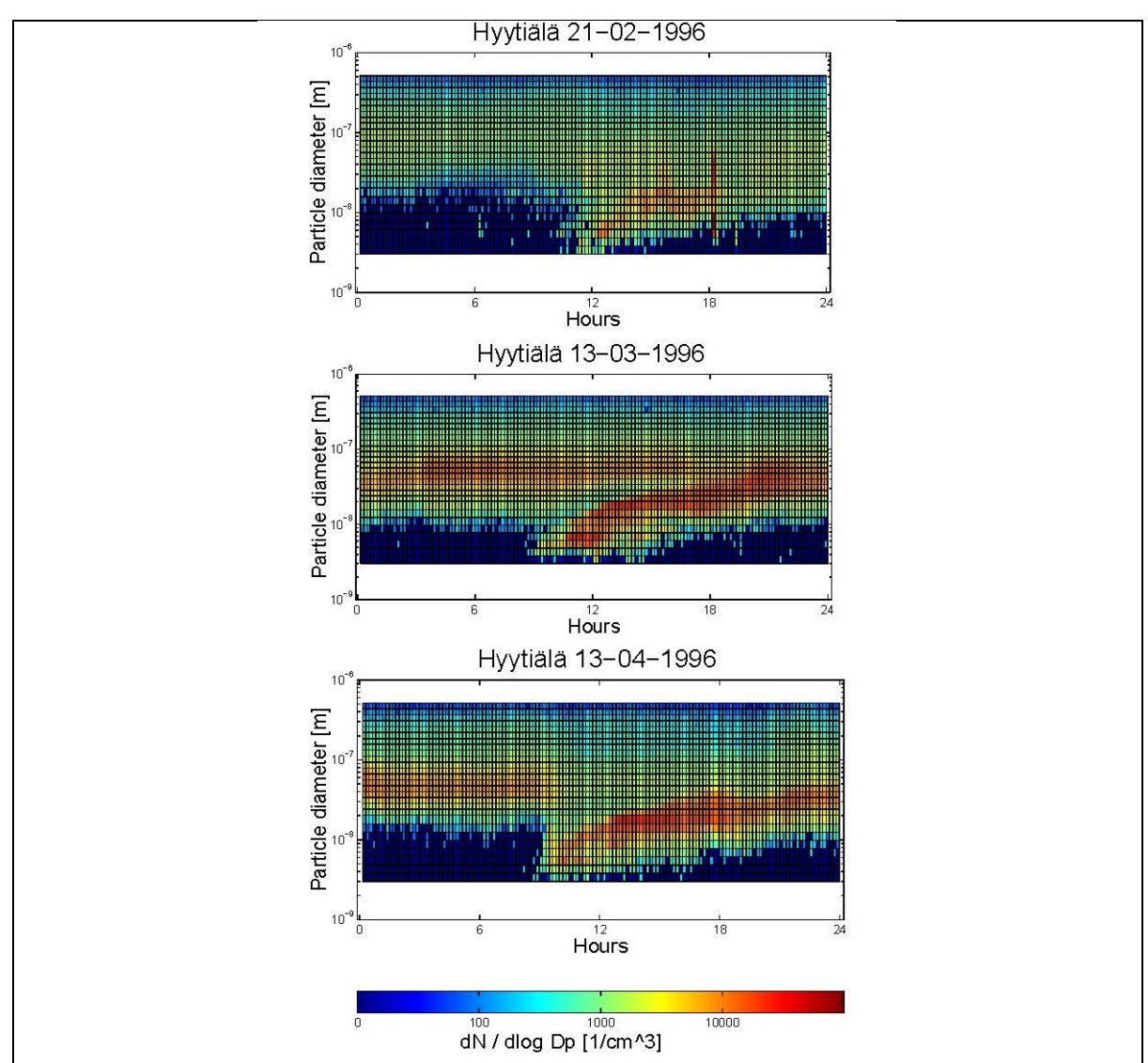

Figure 4 - Contour plots of particle formation event occurring in the morning, followed  by a subsequent growing process of the nucleation mode  during the afternoon. a) February 21st 1996, b) March 13th 1996 and c) April 13th 1996 (Mäkelä et al., 1997).  Thought to be the origin of the "banana plot."


There has been ample recognition for the research on process representations, such as the mole-
fraction-based thermodynamic models (Clegg et al., 1998a, b) and one-parameter model for
hygroscopic growth (Petters and Kreidenweis, 2007) and CCN; Facchini et al. (1999) presented
experimental work aimed at understanding the role of surface tension in droplet growth and the
subsequent effect on cloud albedo and radiative forcing (RF), while Knipping et al. (2000) used a
simplified experiment to investigate the role of reactions of gases with ions at the air-water interface.
More specifically, the role of organics in the formation and growth of aerosol particles has been a
significant area of research (Kulmala et al., 2000). In addition to the natural hydrocarbons noted
above, it has become clear that anthropogenic hydrocarbons such as aromatic compounds are also
involved in new particle formation and their growth (Odum et al. (1997).

Jaenicke (2005) was the first to suggest, to the best of our knowledge, that biological particles are an
important fraction of atmospheric aerosol particles. This paper prompted the development of a new
and exciting field within atmospheric sciences. Fröhlich-Nowoisky et al. (2016) reviewed the role of
bioaerosols in health, climate and ecosystems.

*2.3. Secondary Organic Aerosols*
Since the mid-2000s, secondary organic aerosols (SOA) have been the focus of much research,
addressing their abundance, sources, and production pathways. One of the foundational works in this
area is the recognition of the role of natural and anthropogenic hydrocarbons, and in particular
isoprene chemistry in the formation of SOA (Claeys et al., 2004).

The chemical composition of SOA across the globe is still poorly understood (Zhang et al., 2007),
although ways to describe the growth of SOA have advanced significantly (Kalberer et al., 2004).
Donahue et al. (2006) developed an approach based on the volatility of organics, a concept termed
"volatility basis set." This concept has been extended to a host of volatilities and their classifications.
For example, as shown in Figure 5, Robinson et al. (2007) postulated that a large amount of SOA mass
is unexplained by current models, and methods used to estimate SOA production do not capture what
is measured in the field.

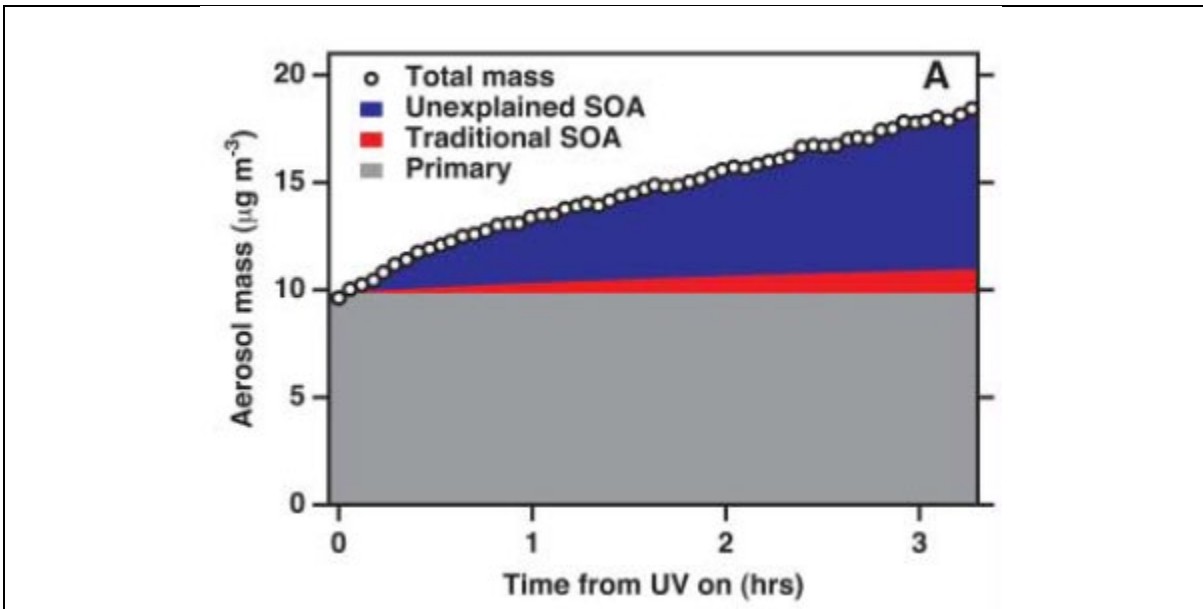

Figure 5 - Aerosol particle formation from the photochemical oxidation of diesel exhaust in an
environmental chamber. The grey area indicates the primary aerosol particle (POA + other species).
The red area shows the upper-bound estimate of the contribution of known SOA precursors to the
suspended aerosol particle mass leaving behind a large fraction that is not accounted for (blue area)
(Robinson et al., 2007).


The introduction of the aerosol mass spectrometer (AMS) by Worsnop and colleagues (Canagaratna
et al., 2007) along with the pioneering instruments of Prather (Gard et al., 1997) and Murphy et al.
(2006) that built on the early work of Sinha (1984), have helped determine aerosol composition.
Studies using these instruments have established that organic compounds are ubiquitous in aerosol
particles. Zhang et al. (2007) and later Jimenez et al. (2009) explored the chemical composition of PM
at different sites across a part of the globe (Figure 6), and their work has now been extended by a
large number of groups.
Aimed at addressing some of the 'missing urban SOA' in models, Surratt et al. (2010) investigated SOA
production from isoprene and Virtanen et al. (2010) showed the amorphous solid state of biogenic
secondary organic aerosol particles, challenging the traditional views of the kinetics and
thermodynamics of SOA formation and transformation, that assumed low viscosity, liquid-like,
particles exchanged chemicals rapidly with the gas-phase.

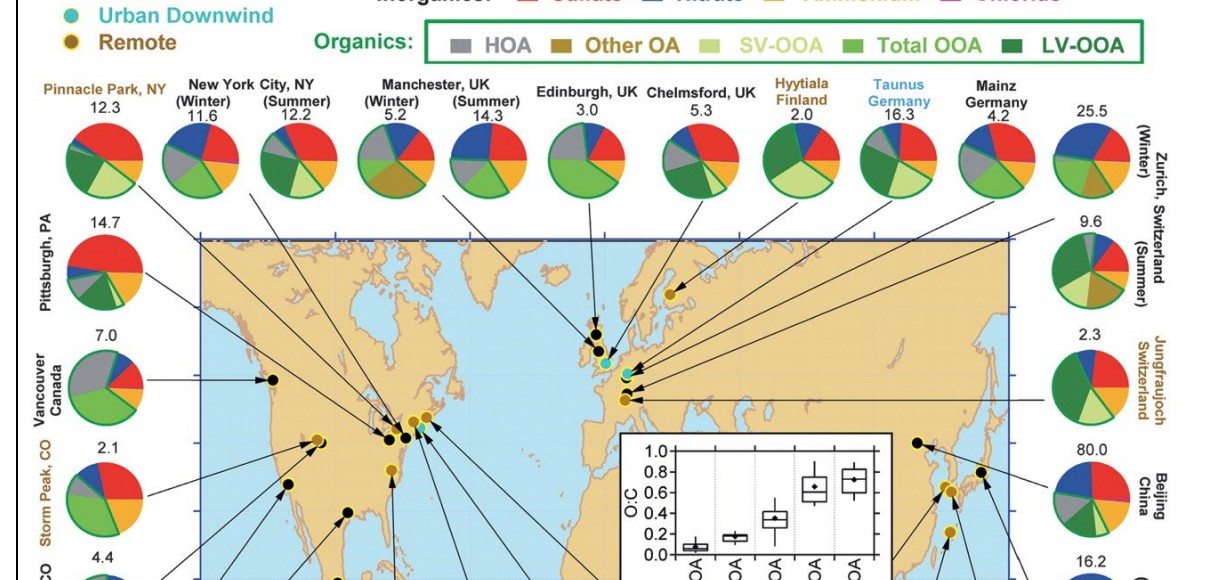

Figure 6 - Total mass concentration (in micrograms per cubic meter) and mass fractions of nonrefractory inorganic species and organic components in submicrometer aerosol particles measured with the AMS at multiple surface locations in the Northern Hemisphere at mid-latitudes. The organic components were obtained with FA-AMS methods (Zhang et al., 2007). In some studies, the FA-AMS methods identified one OOA factor, whereas in other locations, two types, SV-OOA and LV-OOA, were identified. HOA is a surrogate for urban primary OA, and Other OA includes primary OAs other than HOA that have been identified in several studies, including BBOA. Inset: distributions of O:C for the OA components identified at the different sites (Jimenez et al., 2009).


*2.4. Chemical Kinetics, Laboratory Data and Chemical Mechanisms*

Chemical kinetics is one of the foundations of atmospheric chemistry (Abbatt et al. (2014). This includes a number of different areas: investigation of individual chemical reactions; theoretical understanding of elementary reactions; evaluations and compilation of kinetics data; development and compilation of chemical mechanisms for use in models (see the 2.6. *Chemical Models* section); testing and simplification of the models for use in numerical models. Demerjian et al. (1974) is considered by many in the community to be one of the cornerstones of chemical mechanism development and it has been influential in a number of other research areas as well. This paper provided an explicit chemical mechanism for the troposphere in which all the chemical reactions were written as numerically integrated stoichiometric equations to predict photochemical ozone production rates. Previously, all chemical mechanisms had been highly "reduced" (into simple mechanisms) and/or parameterised, with non-stoichiometric equations. Using Demerjian's approach, many explicit atmospheric chemical mechanisms have been derived, including one of the most widely used, the Master Chemical Mechanism (MCM, Figure 7) (Jenkin et al., 1997;Jenkin et al., 2003;Saunders et al., 2003). Currently, there are a variety of tropospheric chemistry mechanisms that capture the scope of chemical reactions that are used in a number of models including the 1990 Carter mechanism (Carter, 1990), the regional acid deposition model / regional atmospheric chemistry mechanism (RADM/RACM) (Stockwell et al., 1997;Stockwell et al., 1990), SAPRC-07 (Carter, 2010) and the Chemical Aqueous Phase Radical Mechanism (CAPRAM) (Ervens et al., 2003). One of the key foundational techniques for estimating rate constants is that of structure-activity relationships (Kwok and Atkinson, 1995). In the near future, calculations of rate coefficients based on ab-initio quantum calculations will likely be common.

There is no doubt that the chemical kinetic data compilations have been the backbone of providing much needed experimental data to all chemical mechanisms and models (see the 2.6. *Chemical Models* section). The comprehensive reviews of Atkinson starting in the mid 80s (Atkinson (1986)) and followed by many others, provided a consistent description of the reaction pathways of the alkyl, peroxy and alkoxy radicals produced by the reactions of hydroxyl radicals with a wide range of organic compounds. These papers led the way for the compilation of the IUPAC and NASA/JPL chemical kinetic data evaluation of tropospheric reactions (Atkinson et al., 1989;Crowley et al., 2010;Atkinson et al., 1992;Atkinson et al., 2004;Atkinson et al., 2006;DeMore et al., 1997;Burkholder et al., 2020). (Note that compilation of kinetics data for stratospheric reactions dates back to mid-1970s (Hudson and Reed, 1979)). These works have been the foundation for the development of all chemical mechanisms and have led to the standardisation and improvement of condensed chemical mechanisms used in all chemical models.

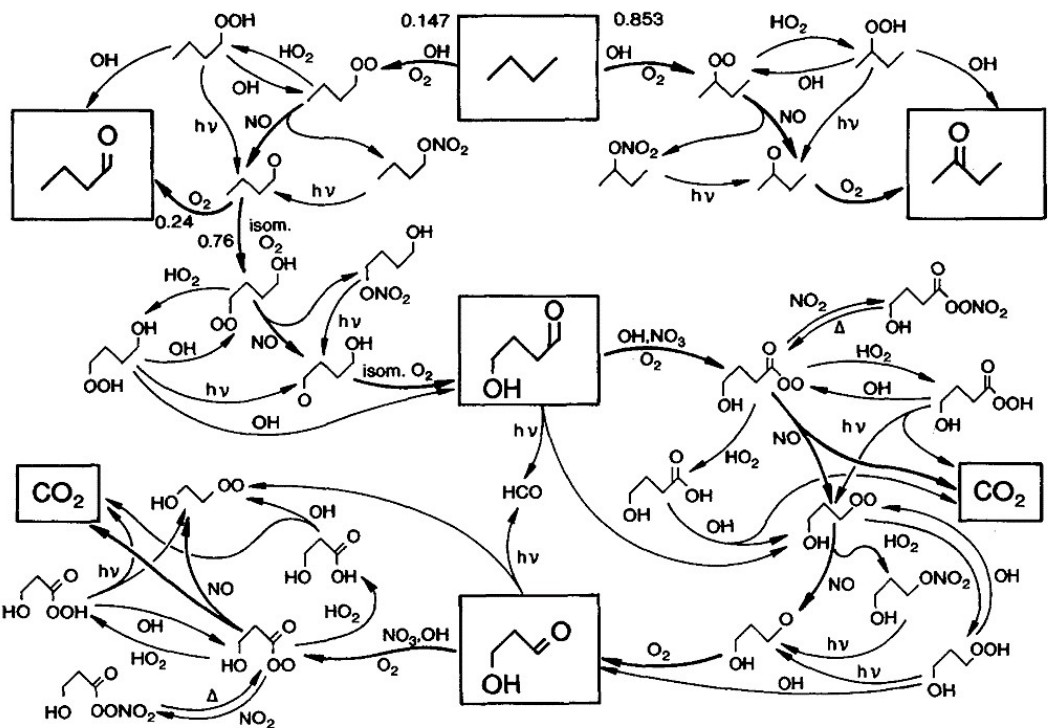


Figure 7. Oxidation mechanism of butane in the Master Chemical Mechanism (Jenkin et al., 1997).

The recognition of reactions with negative activation energies and the role of weakly bound adducts
were other key steps in improving our understanding of chemical kinetics. When the adduct is strong,
we term it an association reaction, which exhibits negative activation energies and pressure
dependence. Understanding and representing these type of reactions in atmospheric chemistry has
been a major step. In particular, the pioneering work of Troe and coworkers has enabled realistic and
simpler representation of these reactions based on the Rice-Ramsperger-Kassel-Marcus (RRKM)
theory (Troe, 1979, 1994).
Atmospheric chemistry is often termed atmospheric photochemistry since the initiator for many of
the reactions is the production of free radicals, which are directly or indirectly the result of solar
radiation. Over the decades, the representation of photochemical processes has been well
established. A key element is the calculation of the "j-value" (i.e. the photolysis rate) for a
photochemical process, which depends on radiative transfer to obtain the solar flux and laboratory
measurements of absorption cross sections and quantum yields. The pioneering works on methods
for quickly and accurately calculating j-values are those of Madronich and Flocke (1999) and of Prather
and colleagues (Wild et al., 2000).
Moving towards individual reactions, the work of Howard and Evenson (1977) on the reaction
between hydroperoxyl radical and nitric oxide, $HO_2$ + NO, has been recognised as a scrupulously
careful study that overturned conventional wisdom on this key reaction in photochemical smog/ozone
formation (and in stratospheric chemistry). The work of Vaghjiani and Ravishankara (1991)
demonstrated the importance of operating at low [OH] to reduce secondary reactions and extended
measurements down to low and atmospherically relevant temperatures.
Three papers nearly a decade apart address the fundamental importance of robust laboratory
measurements to underpin model-led interpretation of experimental data. The seminal work
demonstrating the long-wavelength tail on the ozone photodissociation quantum yield (Ball et al.,
1993;Ravishankara et al., 1998) and the related work on the $O^1D + N_2/O_2$ reactions (Ravishankara et
al., 2002), identified key processes in the formation of OH radicals in the troposphere. Prompted by
the findings of Lelieveld et al. (2008) (see the 2.13. *Biogenic Emissions and Chemistry* section) and of
Hofzumahaus et al. (2009) (see the 2.9. *HOx Chemistry* section), a pair of laboratory papers published
in 2009, about $HO_x$ radical regeneration in the oxidation of isoprene (Peeters et al., 2009) and
unexpected epoxide formation in the gas-phase photooxidation of isoprene (Paulot et al., 2009) have
changed the way we understand the gas and aerosol products and impacts of isoprene chemistry
(Kleindienst, 2009).  It is worth noting that Peeters et al. (2009), using theoretical electronic-structure
calculations, showed the major role of autoxidation chemistry (peroxy – hydroperoxy isomerization),
This work changed the traditional view of peroxy radical chemistry and introduced the ideas of
isomerization and more complex pathways to atmospheric chemistry.

*2.5. Heterogeneous and Multiphase Chemistry*
Earth's atmosphere contains various amounts of condensed matter suspended in air. The most visible
condensed matter is, of course, clouds. One can also see aerosols when the particle numbers and sizes
are large; examples include smog, wildfires, and volcanic eruptions.  In addition to clouds, snow and
ice provide different mediums that can alter gas-phase chemistry.
Many chemical reactions occur on the surfaces of particles suspended in air, ice/snow on the ground,
and within liquid drops. In general, these processes catalyse reactions that would be very slow in the
gas-phase, such as those between closed-shell molecules, and/or can produce products that do not
form in the gas-phase. For these reasons, heterogeneous and multiphase reactions are of immense
interest, although the distinction between heterogeneous and multiphase chemistry is not always
clear cut (Ravishankara, 1997). Often "heterogeneous" is taken to mean reactions at surfaces and
"multiphase" to mean reactions involving the uptake of gases into (and reaction in) the liquid phase.
The unique contribution of reactions in/on condensed matter burst into the limelight owing to their
role in the depletion of stratospheric ozone (Solomon et al., 1986).  However, such reactions had been
recognized to be important before the ozone hole research, for example in the oxidation of $SO_2$ (Urone
and Schroeder (1969) and Penkett et al. (1979)). Since the 1990s, the roles of heterogeneous and
multiphase reactions have been highlighted in many tropospheric processes, as noted here in various
sections (see for instance sections 2.1. *Foundations* and 2.4. *Chemical Kinetics, Laboratory Data and*
*Chemical Mechanisms*).
Chameides and Davis (1982) studied the free radical chemistry of cloud droplets and its impact upon
the composition of rain showing that the radical chemistry in water droplets could drive production
of peroxides, which have the ability to rapidly oxidise sulphur species – a strong link to acid rain. Work
of Akimoto et al. (1987) on the photoenhancement of nitrous acid formation in the surface reaction
of nitrogen dioxide and water vapour demonstrated the existence of an additional radical source in
smog chamber experiments. This built on the earlier work of Pitts et al. (1984) and challenged our
understanding of the role of such heterogeneous reactions in the atmosphere. The work of
Mozurkewich et al. (1987), Hanson et al. (1992) and subsequently Thornton and Abbatt (2005), on the

measurements of $HO_2$ uptake to aqueous aerosol particles was highly influential in the debate on the aerosol loss of $HO_x$, a question that had vexed many modelling studies.

A pioneering paper in tropospheric cloud chemistry is the study by Jacob et al. (1986) in the San Joaquin valley that used a multiphase measurements and modelling approach to study the formation of acid fog. Two further papers have brought heterogeneous chemistry to the fore: Dentener et al. (1996) in their original paper on the role of mineral aerosol as a reactive surface in the global troposphere showed the potential role of mineral dust on sulphur oxides ($SO_x$), $NO_y$ ($NO_y$ includes nitrogen oxide (NO) and nitrogen dioxide ($NO_2$)), as well as the compounds produced from the oxidation of those) and $O_3$ chemistry, and Jacob (2000) who reviewed in more detail the chemistry of ozone *via* $HO_x$ and $NO_y$ at the interface of gas phase and suspended particles (including clouds) and led to a highly-cited series of recommendations for future studies.

Another similar area is that of chemical reactions on/in snow and ice. Such reactions were highlighted by Barrie et al. (1988), specifically with regard to the production of halogens on the ice surface, and hinting at the role of the cryosphere as a source of chemical species to the troposphere (see the 2.11 *Halogen Chemistry* section). Given the extent of the cryosphere and in particular of snow (Grannas et al., 2007), findings in the late 1990s demonstrated its role in promoting heterogeneous and multiphase reactions as a significant source of unusual and unexpected chemical species to the atmosphere. One of the most nominated works in this area was that by Honrath et al. (1999) investigating $NO_x$ production from the illuminated snowpack. Pioneering work of Davis et al. (2001) on the unexpected production of $NO_x$ in pristine Antarctica is also worthy of note.

### 2.6. Chemical Models

Up front, we want to acknowledge that we are not doing full justice to the important role played by chemical models in the understanding of and developing tropospheric chemistry, informing policymaking, and deciphering field observations. We do, however, note some of the key developments in modelling, which is the way we couple atmospheric motion with chemical processes.

Chemical models are the conduit to represent our knowledge of the chemical and physical processes in the atmosphere within a mathematical (numerical) framework that allows prediction and testing against observations (in the laboratory and the atmosphere). Therefore, models are the tools upon which atmospheric environmental policies are developed. Indeed, the efforts in modelling are vast and they are pivotal tool of tropospheric chemistry. Policies pertaining to climate, air quality, acid precipitation, etc. are based on such model predictions and projections. Further, models have been the tool that have enabled quantification of emissions (the quantity of most interest to policymakers), identification of the sources, and evaluation of impacts. One could argue that our knowledge would be incomplete without models.

Early, simple chemical models (with no chemical transport) were useful tools to elucidate and test the basic theory of photochemical ozone formation (Levy, 1971). The recognition that one cannot treat the chemical transformation without considering atmospheric transport and mixing came early. The original simple 1-D models, often designed with a parameterized vertical transport in terms of an "Eddy Diffusion" concept, were superseded by 2-dimensional models and have now been largely replaced by complex 3-D models. 2-D models of the stratosphere (which are zonal averages with

latitude and height being the variable dimensions) have been extremely useful and are still used in assessment activities. (See for example, Garcia and Solomon (1983) and Fleming et al. (1999)). The NASA conference publication 3042 (Jackman et al., 1989) provides an excellent review of sixteen 2-D and a few 3-D stratospheric models that were used in the 1980s and 1990s. Also, chemical transport models (CTMs), which use analysed winds, are often used to separate transport from chemistry. Such models are extremely powerful in accounting for observations, atmospheric budget calculations, and deciphering the roles of various chemical processes taking place in the atmosphere. However, projections and predictions in a changing climate requires coupling of chemistry to meteorological prediction models. Now, free-running, on-line 3-D models, which include chemistry, have been implemented, and the continued enhancements in computing capabilities have greatly improved our modelling capabilities. Logan and co-authors are recognised by many contributors as providing the basic model description of global tropospheric chemistry (Logan et al., 1981). Bey et al. (2001) first described GEOS-Chem, a global, three-dimensional, tropospheric, chemical transport model. Though not the only global tropospheric model, as an open-source model with a large user community and flexibility, it has become a very influential global model. In recent years the Weather Research and Forecasting (WRF)-Chem model has also been used extensively (Grell et al., 2005). There are now numerous 3-D models developed by various organizations. We refer readers to two excellent articles that describe the role of models and their development in detail (Brasseur (2020) and Zhang (2008)). Inverse modelling especially for source and sink attribution has been shown to be a powerful tool (Hein et al., 1997).

In addition to the global models, regional and air-shed models were critical for air quality predictions and are still employed for regulatory use. A series of three papers from the Seinfeld group (Reynolds et al., 1973;Reynolds et al., 1974;Roth et al., 1974) provided the earliest complete descriptions of an air quality policy model. They linked together emission inventories, meteorological data, chemical mechanisms, and air quality network data to evaluate model performance. All subsequent air quality policy models have followed the same general approach and their basic formulation.

Another major use of models is to interpret large scale field measurements. One of the earliest detailed tropospheric chemical modelling studies that integrated highly instrumented intensive field campaign data was that of Harriss (1988) for the ABLE 2A (Amazon Boundary Layer Experiment) campaign in the Amazon Boundary Layer. Now use of multiple models to interpret field data is a common feature of modern tropospheric chemistry research.

Multi-model ensembles of the troposphere as epitomised by Stevenson et al. (2006) and Fiore et al. (2009) (see the 2.16. *Chemical Transport* section) are a powerful tool for generalising the model "understanding" of the atmosphere. This modelling approach makes use of many different models to achieve a more accurate representation of the observations than would be possible by using only one model, thus producing more reliable outcomes for assessments and policies on a global scale. In addition, multiple runs of the same models with slightly different initial conditions are used to examine the range of outcomes. This approach is akin to the use of multiple models and model runs in weather predictions.

*2.7. Tropospheric Ozone*

Ozone is one of the central molecules of atmospheric chemistry and runs through much of the foundations of the discipline, from its role in the stratosphere as a UV shield, to its role as a major greenhouse gas, to its pivotal part in the troposphere as the start and end-product of oxidation chemistry, and its detrimental influence as an air pollutant harmful to human health and ecosystems. Much of the early thinking on ozone was focused on the question of whether tropospheric ozone was a small subset of stratospheric ozone, see for example Galbally (1968) and Fabian and Pruchniewicz (1977). The latter paper showed the value of observational networks based on standardised instrumentation and calibration techniques, together with consistent siting criteria, and raised the issue of seasonal variations in tropospheric ozone and the nature of the processes that drive them. The vertical structure of a layer of high $O_3$ concentrations in the stratosphere, where $O_2$ could be directly photolyzed to make oxygen atoms and hence ozone, and declining concentration in the troposphere was indicative of a stratospheric source and a tropospheric sink and this was the prevalent theory prior to the late 1970s (see also the 2.18. *Stratospheric Chemistry* section). A major breakthrough were the two papers by Chameides and Walker, and Crutzen (Chameides and Walker, 1973;Crutzen, 1973b) that showed that ozone can be photochemically generated in the troposphere, just like it is made in smog *via* the reactions involving hydrocarbons and nitrogen oxides.

The importance of ozone as a radiative gas has been known for a long time, with a significant fraction of heating in the stratosphere coming from ozone photolysis followed by its reformation and thus converting sunlight to heat. In 1979, Fishman et al. (Fishman et al., 1979) identified that tropospheric ozone is also a greenhouse gas. Hence, a change in tropospheric ozone will perturb the radiative energy budget of the Earth-atmosphere system which will in turn perturb the climate system. Ozone thus became the second trace gas after carbon dioxide to be implicated in global warming and climate change.

Large scale mapping of global tropospheric ozone was first undertaken by Logan (Logan, 1985) who looked at seasonal behaviour and trends with a view to understanding anthropogenic influence. This was later complemented by a paper exploring the photochemical origins of tropospheric, rather than stratospheric, ozone in the rural United States (Logan, 1989). As the understanding of the photochemistry of ozone developed, measurements at Niwot Ridge, Colorado (Liu, 1987) aimed to quantify the elements of the ozone budget by season, bringing forward the concept of ozone production efficiency. Lin et al. (1988) explored the non-linearity of tropospheric ozone production with respect to NMHCs and NOx. Though this chemistry had been outlined much earlier – e.g. Demerjian et al. (1974) – this work explored it in the background atmosphere with models and measurements. A powerful demonstration of the low-NOx ozone destruction chemistry came from measurements made at Cape Grim, Australia, a background station, where Ayers and Penkett (Ayers et al., 1992) and their team(s) used measurements of ozone and peroxides (Figure 8) to show further experimental proof for the photochemical control of ozone in remote locations.

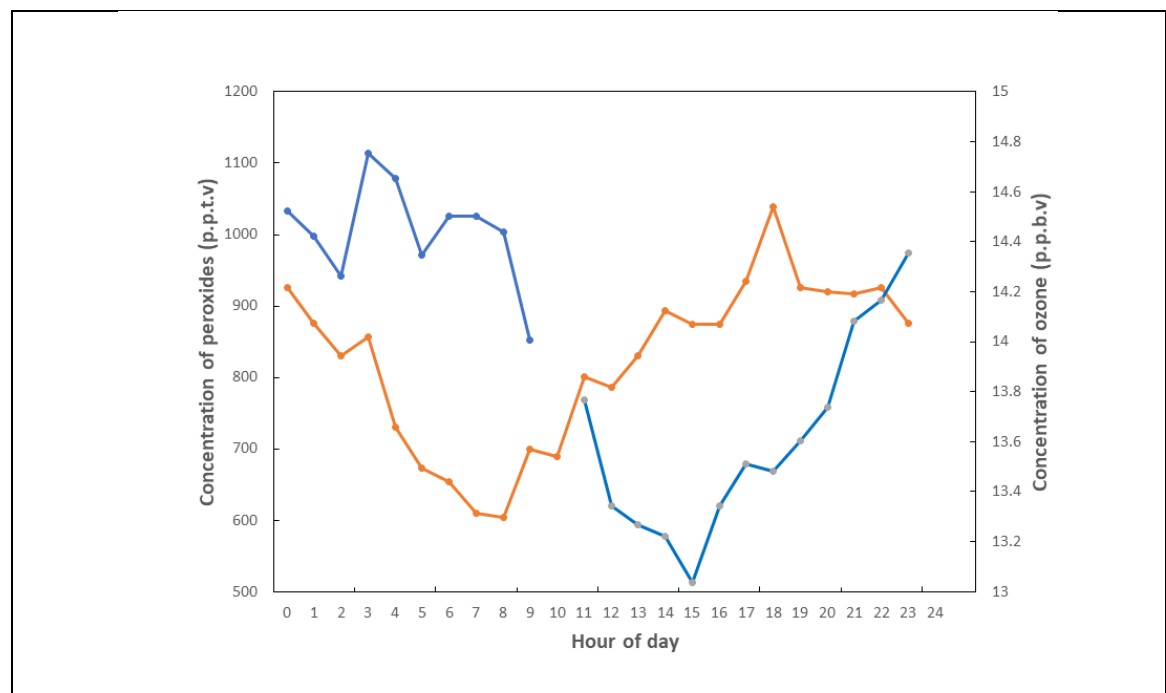

Figure 8 – Average diurnal cycles of peroxide (orange line) and ozone (blue lime) in baseline (low NOx) air at Cape Grim (Tasmania, Australia) in January 1992. Adapted from Ayers et al. (1992).

Measurements have always been a critical driver in tropospheric chemistry, and the idea to use in-service commercial aircraft as a platform for programs such as MOZAIC/IAGOS (Thouret et al., 1998) has been recognised for the enormous amount of high quality data, which would otherwise be difficult to regularly obtain from the upper troposphere and lower stratosphere. Using such measurements Newell et al. (1999) combined dynamical and chemical tracers to further delineate ozone origin and budgets. In the same year, Logan published a synthesis of ozone sonde data (Logan, 1999) which gave an unprecedented look at the seasonal and vertical distribution of ozone and became a reference point for the subject. A year later, Thompson et al. (2000) used a combination of shipboard and satellite views of a tropospheric ozone maximum to suggest the occurrence of a tropical Atlantic ozone "paradox". The "Atlantic paradox" refers to a greater tropospheric ozone column amount over the South Atlantic than the North Atlantic during the West African biomass burning season. This phenomena was further explored using an expanded network of ozonesondes in the southern hemisphere (SHADOZ) (Thompson et al., 2003). In combination with the earlier work of Logan, these became the basis for the measurement-based description of ozone in the troposphere.

A decades worth of knowledge on the relationship between ozone and its precursors was pulled together by Sillman (Sillman, 1999), cementing the concepts of NOx- and VOC-sensitive (or NOx saturated) chemical regimes. The paper introduced a generation of researchers to isopleth diagrams (the famous Sillman plot Figure 9) and ozone production efficiencies (OPEs).

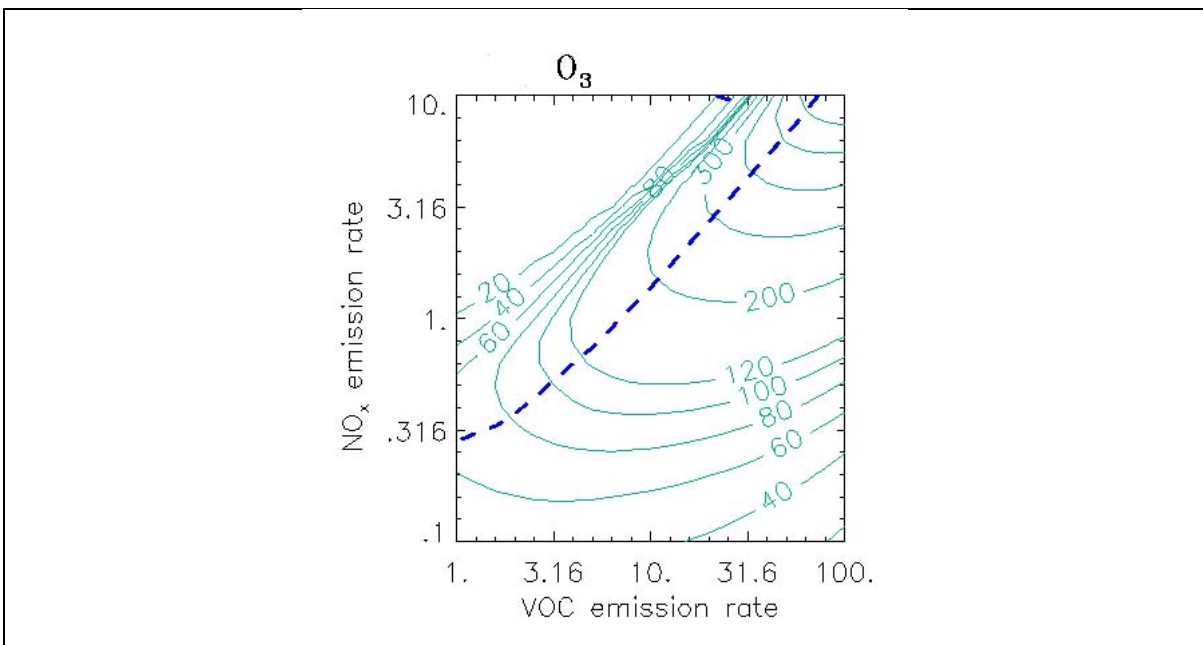

Figure 9 - . Ozone isopleths (ppb) as a function of the average emission rate for $NO_x$ and VOC (in $10^{12}$ molec. $cm^{-2}s^{-1}$) in 0-D calculations. The isopleths (solid green lines) represent conditions during the afternoon following 3-day calculations with a constant emission rate, at the hour corresponding to maximum $O_3$. The short blue dashed line represents the transition from VOC-sensitive to $NO_x$-sensitive conditions. Adapted from Sillman and He (2002).


The power of models to explore global tropospheric ozone distributions, budgets and radiative forcing
was fully demonstrated in the study by Stevenson et al. (2006), that brought together 26 atmospheric
chemistry models to explore both the air quality and the climate roles of ozone (see also the 2.6.
*Chemical Models* section). As discussed in the 2.16. *Chemical Transport* section, a similar approach
was used by Fiore et al. (2009) to explore the relationship between inter-continental transport and
ozone.

### 2.8. Nitrogen Chemistry
Nitrogen oxides are an integral part of tropospheric processes. Nitrogen oxides are released into the
troposphere from a variety of biogenic and anthropogenic sources including fossil-fuel combustion,
biomass burning, microbial activity in soils and lightning. The concept of the Leighton photostationary
state (Leighton, 1961) between NO, $NO_2$, and $O_3$ was well established by the mid-1990s, and early
work from Singh and Hanst (1981) highlighted the potential role of peroxyacetyl nitrate (PAN) to be a
reservoir for $NO_x$ in the unpolluted atmosphere.  The measurement of $NO/NO_2$ by chemiluminesence
was critical to the widespread measurement of NOx (Kley and McFarland, 1980).   A landmark paper
in the area of nitrogen chemistry was that of Logan (1983) that brought together global and regional
budgets for the nitrogen oxides (Table 3). Later, a paper that focused more narrowly on a specific
source of $NO_x$ was that of Yienger and Levy II (1995) who produced an empirical model of global soil-
biogenic $NO_x$ emissions. Higher up in the atmosphere, the work on sources and chemistry of $NO_x$ by
Jaeglé et al. (1998) is recognised for its contribution to the understanding of the $NO_x$ cycle in the upper
troposphere.
HONO, somewhat a Cinderella molecule, whose photolysis can be a major OH source, especially during
the early morning was first identified by Perner and Platt (1979), the heterogeneous nature of which
has always driven much interest (Kurtenbach et al., 2001).
These works were complemented by a more holistic view of the nitrogen cycle and in particular the
concept of reactive nitrogen (Nr) by Galloway et al. (2004) that clearly showed the linkages between
the terrestrial ecosystem and the atmosphere and how the nitrogen budget had and would change
leading to the important concept of nitrogen cascade (Sutton et al., 2011). In more recent times,
extensive work on vehicle NOx sources from exhaust remote sensing data (Bishop and Stedman,
1996), as epitomised in Carslaw (2005) should be highlighted. This paper pointed out the trends that
can be said to have led to the denouement of the Volkswagen emissions scandal.

*2.9. HOx Chemistry*
There is no doubt that the chemistry of OH and $HO_2$ (known together as $HO_x$) has a central role in the
atmosphere as well as holding a certain fascination to atmospheric scientists owing to the significant
challenges involved in measurement and understanding its impact locally to globally. Much of the
history of the measurements of OH and $HO_2$ is covered in the review of Heard and Pilling (2003). As
they wrote "clearly, OH plays a central role in tropospheric chemistry. The in situ measurement of its
concentration has long been a goal, but its short lifetime and consequently low concentration provide
a serious challenge."
In order to assess the global impact of OH chemistry in the absence of direct measurements, reactive
proxies have been used.  Singh (1977) used methyl chloroform to estimate OH abundance since methyl
chloroform is exclusively anthropogenic and were emissions are known. This type of work provided a
comprehensive picture of the global distribution of OH and, hence, a first overall look at the oxidative
capacity of the atmosphere. It was followed, using halocarbon measurements by the AGAGE network,
by a global OH determination, while also introducing the atmospheric chemistry community to formal
inverse modelling (Prinn et al., 1995). Spivakovsky and co-workers expanded on this work to derive 3-
D distributions of OH and used this information to assess the wider impact on the lifetimes of
halocarbons, which have implications for stratospheric ozone (Spivakovsky et al., 2000). Thanks to the
availability of long term observations of halocarbons from the AGAGE and NOAA networks, later work
using a similar approach found evidence for substantial variations of atmospheric hydroxyl radicals in
the previous two decades (Prinn et al., 2001), thus providing a broad overview not only of the global
distribution but also of the temporal variability of this crucial species. Such estimates allowed for the
quantification of the lifetime of important chemicals such as methane and CFC-substitutes such as the
hydrochlorofluorocarbons (HCFCs) and hydrofluorocarbons (HFCs).
The in-situ OH detection in the troposphere has proven elusive for a long time. The use of laser-
induced fluorescence provided some of the first clues to its atmosphere concentrations in the 1970s
and early 1980s (Davis et al., 1976;Wang et al., 1975), but many of these early measurements were
found to have significant artefacts. Long-path UV absorption in Germany showed the OH abundances
in German boundary layer to be around 1-4 x $10^6$ cm$^{-3}$ (Perner et al., 1976b). The study by Eisele (1994)
at the Fritz Peak Observatory in Colorado, was the first intercomparison experiment of different
measurement techniques and provided much needed confidence in the observations of this key
molecule. Stevens et al. (1994) developed the low-pressure laser-induced-fluorescence (LIF)

instrument, which quickly became one of the most successful and widely used techniques for ambient measurements of OH and $HO_2$. As ambient observations of HOx became available, they were found useful to test our understanding of tropospheric chemical processes, by comparing them with the results of chemical models (see the 2.6. *Chemical Models* section). Recognised as a foundational paper in this area, Ehhalt (1999) explained with clarity the role of radicals in tropospheric oxidation and what controls their concentrations, using both ambient measurements and calculated concentrations of OH. The OH radical is particularly suited to test our understanding of chemical processes and this was clearly demonstrated in 2009, when the discrepancies between observed and calculated OH and $HO_2$ in the polluted region of Southern China led Hofzumahaus and co-workers to propose a regeneration pathway for OH, which does not involve NOx and thus does not produce $O_3$ (Hofzumahaus et al., 2009). This, together with the work by Lelieveld and co-workers (Lelieveld et al. 2008), prompted a major reassessment of the isoprene oxidation mechanism by Peeters et al. (2009) who suggested that isomerisation of hydroxyperoxy radicals from isoprene oxidation could be fast enough to regenerate HOx in highly forested, low NOx environments (see the 2.4. *Chemical Kinetics, Laboratory Data and Chemical Mechanisms* section) and led to a major revision of isoprene chemistry and its role in the troposphere (see the 2.13. *Biogenic Emissions and Chemistry* section).

The sources and sinks of HOx radicals have always been a major research focus (Finlayson and Pitts, 1976) and the work of Paulson and Orlando (1996) on the reactions of ozone with alkenes as a source of HOx in the boundary layer is widely recognised. Radical chemistry is highly sensitive to the levels of NO and $NO_2$ and Kleinman's modelling work on hydrogen peroxide ($H_2O_2$) concentrations in the boundary layer is recognised for its simple elegance in describing how the $HO_x$ cycle chemistry is influenced by NOx and in giving insight into the differing fates of OH and $HO_2$ radicals under different NOx regimes (Kleinman, 1991) (Figure 10).

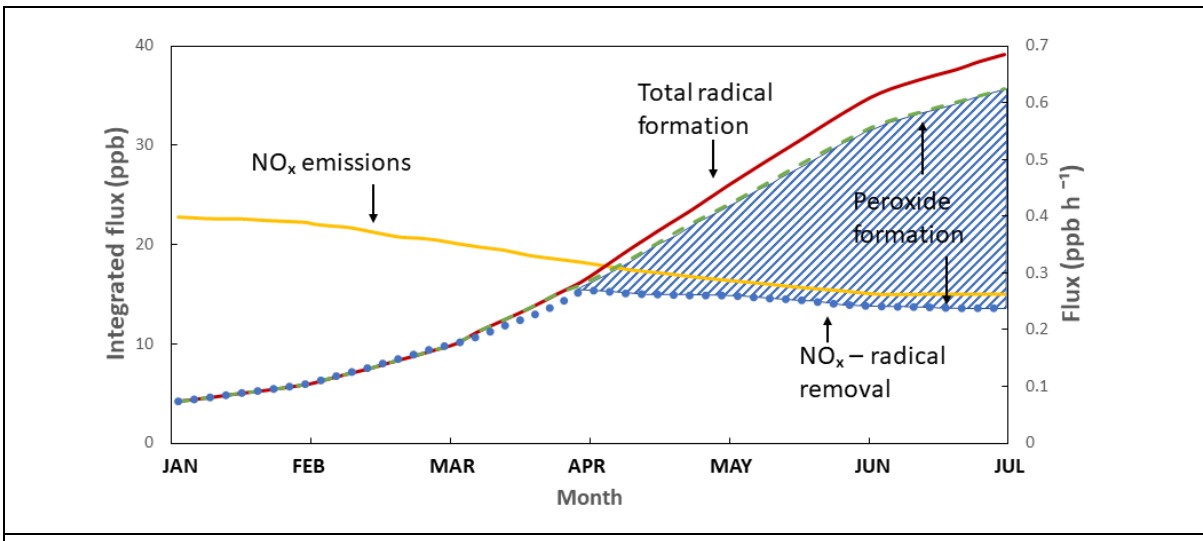

Figure 10 – Calculated seasonal variation of major radical loss processes after Kleinman (1991) showing the seasonal nature of peroxide production and its dependency on NOx and radical formation.

The first direct measurements of OH lifetime (Di Carlo et al., 2004) provided evidence of missing reactivity, i.e., that not all sinks of the OH radical are known, which relates to earlier work by Lewis et

al. (2000) on unmeasured volatile organic compounds (see the 2.12. *Volatile Organic Compounds*
section).

2.10.    *Nightime Chemistry*
There is widespread recognition that the atmosphere's oxidative chemistry is active during the night
as well as during the day. Evidence of nighttime chemistry driven by the nitrate radical ($NO_3$) and
ozone was first observed in the (polluted) troposphere in 1980 by Platt and co-workers (Platt et al.,
1980). Much of the early $NO_3$ work, including laboratory and field studies, is summarised in Wayne's
seminal review (Wayne et al., 1991). Platt and colleagues and Plane and colleagues' ground-breaking
work based on long-path absorption have shown the importance of $NO_3$ in the troposphere (Allan et
al., 1999;Platt et al., 1979).
Two papers that have been highly influential in shaping our view of nocturnal chemistry are "Nitrogen
oxides in the nocturnal boundary layer: Simultaneous in situ measurements of $NO_3$" (Brown et al.,
2003) and "Variability in Nocturnal Nitrogen Oxide Processing and Its Role in Regional Air Quality"
(Brown et al., 2006). Both these papers showed the power of state of the art measurements coupled
with models to assess the impact of nocturnal and heterogeneous chemistry on regional air quality.
In particular, the paper by Brown et al. (2006) was a powerful demonstration of the role of
heterogeneous chemistry and aerosol particle composition in controlling dinitrogen pentoxide ($N_2O_5$)
and, therefore $NO_3$, concentrations (Figure 11).

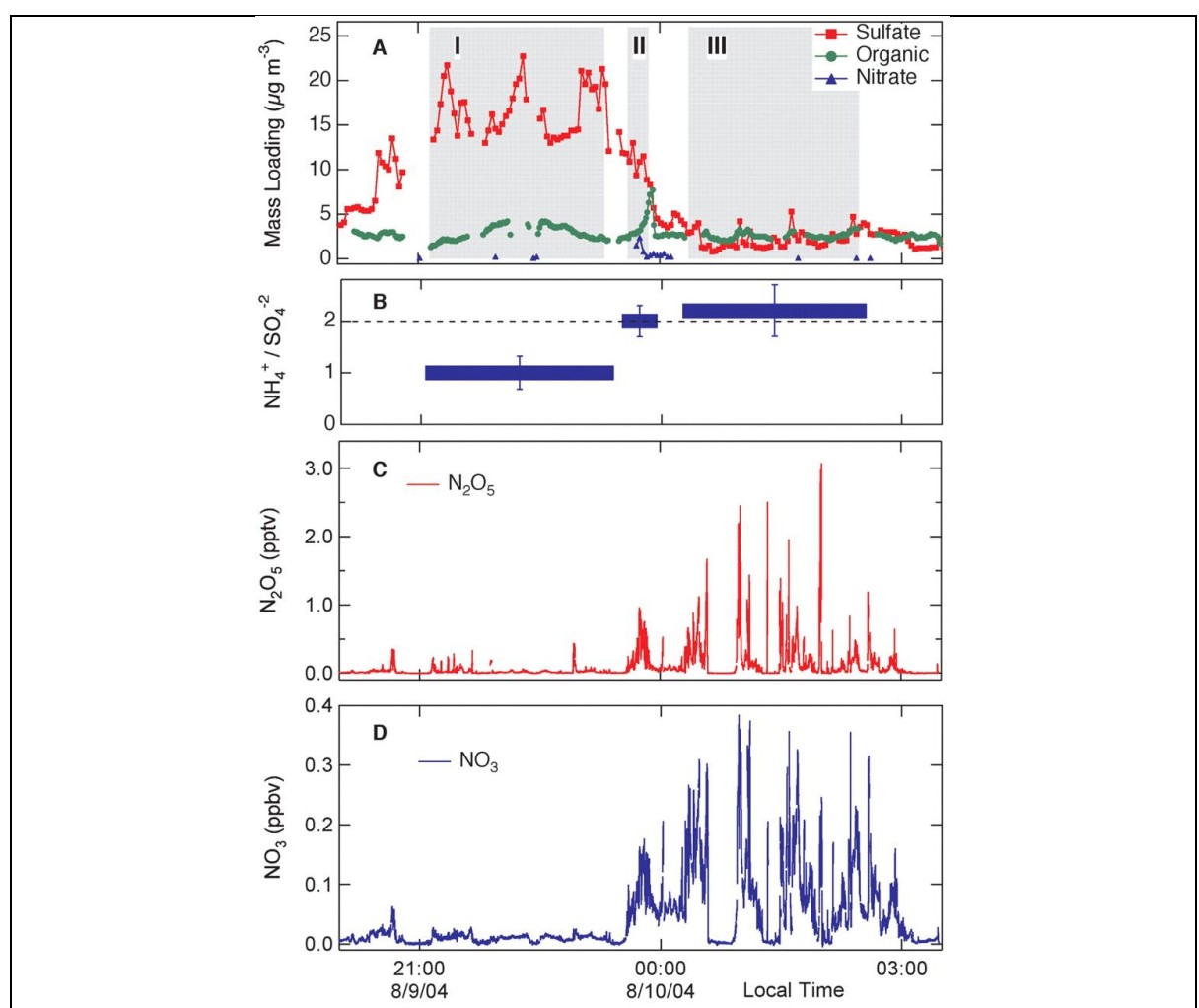

Figure 11 – Measurements during a flight of the NOAA research aircraft P3 for 9 and 10 August 2004, showing the relationship between $NO_3$ and $N_2O_5$ concentration and aerosol particle composition (Brown et al., 2006).


The area of $NO_3$ chemistry is very active and there have been significant further studies since Brown
et al. work. Another area of particular note for nighttime processes are those due to the Criegee
intermediate. The role of Criegee intermediates have been known for a while. However, recent ability
to isolate and measure the reactivity of this intermediate is showing the importance of this radical.

2.11.    *Halogen Chemistry*
In comparison to the atmospheric chemistry in the stratosphere, where halogen chemistry has been
well known and characterized for a long time (see the 2.18. *Stratospheric Chemistry* section), the
recognition of the role of halogen species in the oxidative chemistry of the troposphere occurred much
later. Reviews of the earlier work can be found in Cicerone (1981), Platt and Hönninger (2003), Monks
(2005) and the extensive review by von Glasow and Crutzen (von Glasow and Crutzen, 2007).
The role of halogens in the troposphere has been discussed going back to the 1970s (e.g. (Graedel,
1979)). The potential importance of iodine in the troposphere was highlighted by a seminal paper by
Chameides and Davis in 1980 (Chameides and Davis, 1980). An important early paper is that from
Barrie in 1988 (Barrie et al., 1988) that demonstrated the dramatic impact of bromine chemistry on
Arctic boundary layer ozone (Figure 12). The occurrence of ozone depletion events in the polar
boundary layer suggested that halogens could have a significant impact on atmospheric chemistry at
low altitudes and not just in the stratosphere. This work brought together halogen and heterogeneous
chemistry and led to the discovery of bromine catalyzed ozone depletion on ice-covered surfaces (see
2.5. *Heterogeneous and Multiphase Chemistry*).

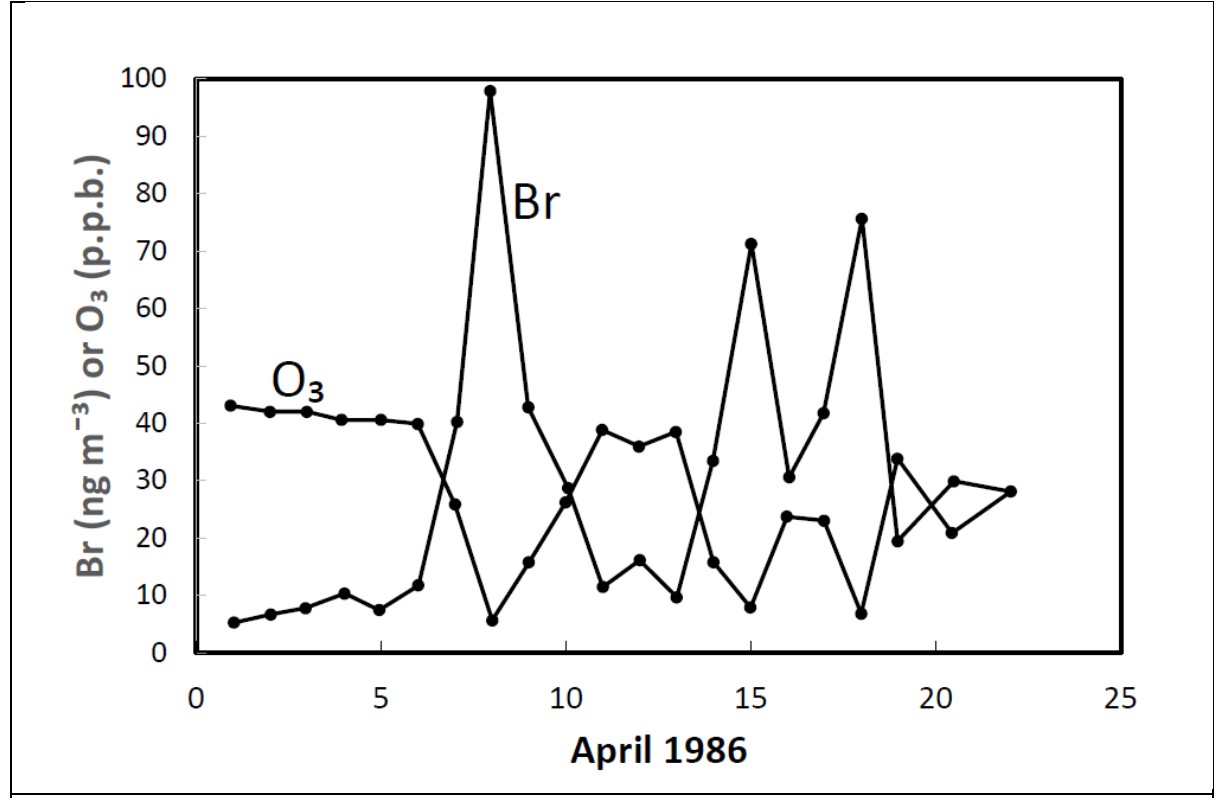

**Figure 12** - A comparison of daily mean ground level $O_3$ and filterable $Br^-$ concentrations at Alert, Canada, in April 1986 illustrating the strong inverse correlation between the two chemicals (Barrie et al., 1988).


One of the foundational papers in the area of halogen chemistry is the modelling study by Vogt et al.
(1996) which set the theoretical framework for the sea-salt activation mechanism for halogen release
and linked halogen chemistry with the sulphur cycle. While the initial research focus was on chlorine
and bromine, Alicke et al. (1999) reported the first iodine oxide observations in the marine boundary
layer at Mace Head, Ireland, and proved that iodine can also be an important player in the chemistry
of the troposphere. Further investigation found evidence that biogenic iodine species can be
responsible for the formation of marine aerosol and cloud condensation nuclei (O'Dowd et al., 2002)
recognizing the potential for wide scale impact of iodine chemistry in particle formation. Finlayson-
Pitts and her colleagues had suggested the importance of chlorine in tropospheric chemistry based
on laboratory data (Finlayson-Pitts et al., 1989) as has been seen earlier by Schroeder and Urone
(1974), but it wasn't until 2008 that Osthoff and co-workers (Osthoff et al., 2008) – and the related
comment "When air pollution meets sea salt" by von Glasow (2008) – brought attention to the
potential for nitryl chloride ($ClNO_2$) chemistry to impact ozone formation, nitrogen recycling and VOC
oxidation, with the first ambient observations of this molecule. Also in 2008, the work from Read et
al. (2008) clearly showed the global importance of halogens for tropospheric ozone using long-term
observations of iodine and bromine oxides (IO, BrO) made at the Cape Verde Atmospheric
Observatory.  High concentrations of tropospheric $Cl_2$ first reported by Spicer et al. (1998) have
been found in other places.

*2.12.        Volatile Organic Compounds*
Volatile organic compounds (VOCs) embraces a wide variety of species emitted from man-made and
natural sources. In many respects VOCs are the fuel of the oxidative chemistry in the atmosphere,
involved in many gas- and particle-phase processes.
Ehhalt (1974) brought together the details of the methane sources and sinks and put them into a
consistent framework that described the life cycle of methane. This conceptual framework has
subsequently been expanded to a wide range of trace organic gases. The original understanding of the
life cycle of methane has remained largely unchanged over the subsequent 40 years and has formed
the basis of the IPCC science assessments on the role of methane in global warming and climate
change. Methane itself has been long recognised as important for tropospheric chemistry, but also for
climate change as a greenhouse gas and as a source of water vapour to the stratosphere. Specifically,
the work of Blake and Rowland (1986) documented the global increase in methane and its implications
for climate change.
The large differences in reactivity among the individual VOCs have always been a feature of their
chemistry. Darnall et al. (1976) produced a reactivity scale for atmospheric hydrocarbons[2] based on
their reaction with the hydroxyl radical, an idea that is still influential to the present day. The concept
was further advanced by Carter and Atkinson (1989) who looked at incremental hydrocarbon
reactivity, where knowledge of the reactivities of organics with respect to ozone formation in the
atmosphere can provide a useful basis for developing appropriate control strategies to reduce
ambient ozone levels. It was the beginning of an approach that is often now used in regulation to
determine which organic compounds would have the greatest effect in reducing ozone.
VOC transformation can be important in a number of different atmospheric processes. One highly
cited early example is the work of Pitts et al. (1978) on the atmospheric reactions of polycyclic
aromatic hydrocarbons and their ability to form mutagenic nitro derivatives under typical atmospheric
conditions.
While measurement techniques rarely seem to get a mention as being influential, the discipline relies
on observations as a critical part of the oeuvre. Already mentioned was the huge impact that accurate
techniques to measure the OH radicals had on the development of the field (see 2.9. *HOx chemistry*).
Another example is the development of Proton-Transfer Reaction Mass-Spectrometry, which has
revolutionised the measurement, in particular, of VOCs (Lindinger et al., 1998) and the earlier work of
Lovelock and Lipsky (1960) on the development and application of electron capture detectors that
allowed the measurement of VOCs such as dimethyl sulphide and the halocarbons in the troposphere.

---

[2] Hydrocarbons, although often used interchangeably with VOC, do not describe the same group of
compounds. Hydrocarbons are a subset of VOC that exclusively contain hydrogen and carbon, and thereby
include none of the e.g., oxygenated VOC species.

Research is ongoing as to how many VOCs there are in the atmosphere and what the consequences
are of not being able to measure/quantify them all. The work of Lewis et al. (2000) used novel VOC
measurements (GC x GC) to find that there was a larger pool of ozone-forming carbon compounds in
urban atmospheres than previously posited (Figure 13). The later paper by Goldstein and Galbally
(2007) expanded on this work hypothesizing that thousands of VOCs are still unmeasured and
unknown, with potentially huge consequences for the carbon budget of the atmosphere.
Continuing work in this area, de Gouw et al. (2005) produced a landmark study that combined analysis
of organic carbon in both the gas- and particle-phase in the polluted atmosphere as part of the New
England Air Quality study by looking at the evolution of VOCs from their emission sources. The study
showed that most of the organic carbon in the particle-phase was formed by secondary anthropogenic
processes and that an increasing fraction of the total organic mass was constituted of oxygenated
VOCs as a result of the air masses being processed/aged.

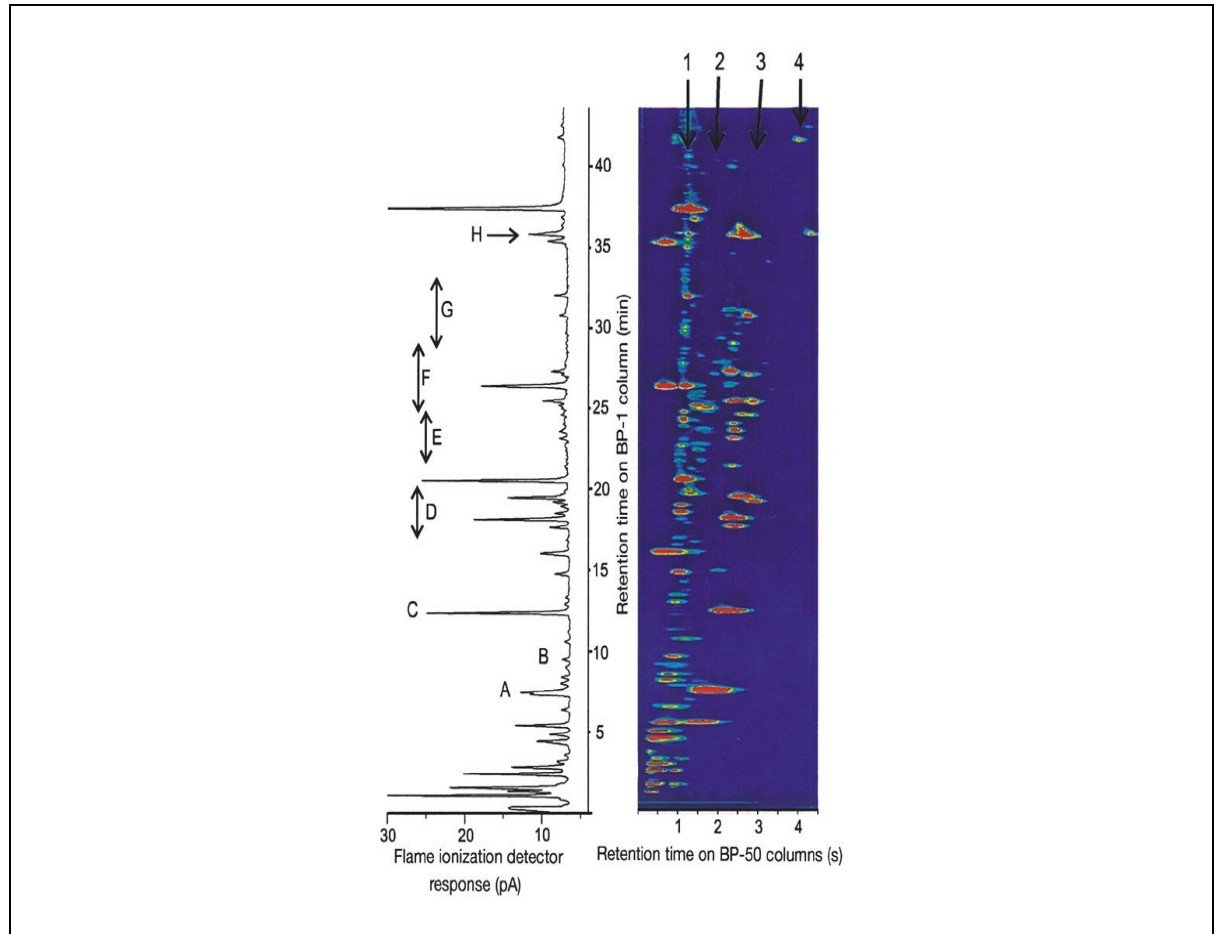

Figure 13 - Comprehensive (right) and one-dimensional (left) separations of volatile organic
compounds in urban air demonstrating a greater variety and number of VOCs (Lewis et al., 2000)


*2.13.        Biogenic Emissions and Chemistry*
Although it has been known for a long time that plants emit organic compounds, the relevance of
biogenic VOCs for atmospheric chemical processes was not immediately recognized.  The first report

that plants emit volatile organic compounds into the atmosphere was made in 1957 by the Georgian scientist Guivi Sanadze (Sanadze, 1957). Unaware of Sanadze's work in the USSR, Rasmussen and Went independently discovered isoprene emissions in 1964 (Rasmussen and Went, 1964). Sanadze was also the first to show that isoprene emission rates are temperature dependent (Sanadze and Kursanov, 1966). However, the relevance of biogenic VOC for atmospheric chemical processes was not immediately recognized.  Went (1960) hypothesised that "volatilisation of terpenes and other plant products results in the production of, first, blue haze, then veil clouds ... ".  Although Tingey (Tingey et al., 1979) at the US Environmental Protection Agency did note the potential for isoprene to play a role in regional air quality in 1978, this was not formalised until the ground-breaking work of Chameides and colleagues in 1988 (Chameides et al., 1988) and MacKenzie and colleagues in 1991 (MacKenzie et al., 1991).

In 1992 the seminal review of Fehsenfeld et al. (1992) brought the importance of isoprene and a wide range of other VOCs of biological origin to the attention of the atmospheric chemistry community, opening up an entirely new branch of atmospheric chemistry. Other influential reviews of biogenic VOC emissions, include e.g., the physiology of plants (Kesselmeier and Staudt, 1999) and more recently Sharkey and Monson (2017) reviewed the enigmatic nature of isoprene emissions.

Over time, biogenic chemistry  became pivotal for major policy formulations to abate ozone pollution. Underpinning the atmospheric chemistry research that Fehsenfeld et al. (1992) promoted, plant physiologists began working on understanding the biological and environmental controls on biogenic VOC emission rates. This work allowed the development of relatively simple functions to predict the emissions of biogenic VOCs which resulted in the first spatially and temporally resolved global model of biogenic emissions (Guenther et al., 1993). These soon evolved into more sophisticated high resolution global models (Guenther et al., 2000;Guenther et al., 1995), allowing for the emissions of biogenic compounds to be included in atmospheric chemistry models across all scales. Eventually, this work took the form of the widely used MEGAN (Model of Emissions of Gases and Aerosols from Nature) model (Guenther et al., 2006) which is still used in modern Earth system models today (Table 1 and Figure 14).

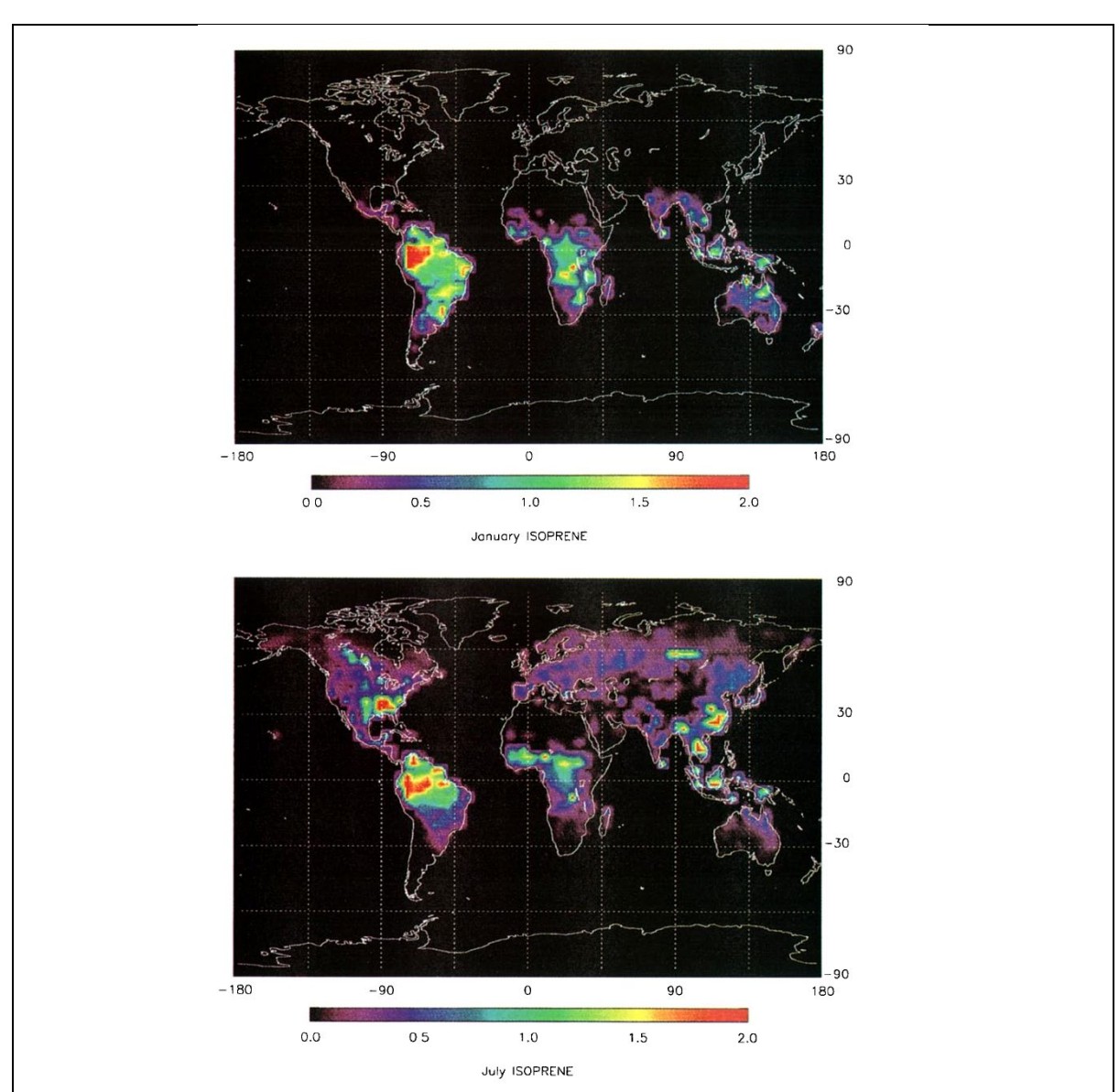

**Figure 14** - Global distribution of isoprene emission rate estimates (g C m$^{-2}$ month$^{-1}$) for (top) January and (bottom) July (from Guenther et al. (1995)).


Terrestrial vegetation is not the only source of biogenic emissions. Aneja et al. (1979) discussed the
importance of biogenic sulphur compounds and their role in stratospheric chemistry, while Charlson
et al. (1987) connected marine biology, atmospheric chemistry and climate into the already
mentioned CLAW hypothesis (see the 2.2. *Aerosols and Clouds* section).
Marine aerosol formation was thought for a long time to be dominated by inorganic components,
mainly sea-salt and non-sea salt sulphate, but O'Dowd and co-workers (O'Dowd et al., 2004) showed
that biological activity of plankton blooms can enhance the concentration of cloud condensation
nuclei, a key aspect of the chemistry-climate feedback mechanism. A similar mechanism is also active
in relation to biogenic halogen compounds (O'Dowd et al., 2002) which also affects aerosol formation
as well as the ozone, nitrogen, and sulphur cycles, as discussed in section 2.2. *Aerosols and Clouds*.
A paper that has been defined as "controversial but set off a huge amount of activity" is "Atmospheric
oxidation capacity sustained by a tropical forest" by Lelieveld et al. (2008), which proposed a new

chemical mechanism for low NOx, high VOC regions (such as tropical forests), based on modelling studies of a field dataset. Although further studies contradicted this hypothesis, Lelieveld et al. (2008) was instrumental in prompting a large amount of laboratory, theoretical and field studies in the past 10 years. These studies resulted in a major revision of our understanding of biogenic VOC chemistry (see the sections 2.4. *Chemical Kinetics, Laboratory Data and Chemical Mechanisms* and 2.9. *HOx chemistry*).

### 2.14. *Biomass Burning*

Biomass burning, particularly in the tropics, affects terrestrial vegetation dynamics, soil erosion, movement of organic carbon, hemispheric atmospheric composition, air quality and more broadly radiative forcing *via* emissions of trace gases and aerosols (Monks et al., 2009). Crutzen (Crutzen et al., 1979) was the first to highlight biomass burning in the tropics as an important source of atmospheric gases, such as molecular hydrogen ($H_2$), CO, $N_2O$, NO, chloromethane ($CH_3Cl$) and carbonyl sulphide (COS). The importance of biomass burning, based on the observations of a small set of fires, and the appreciation of its potential role was a major step in our understanding of the role of biomass burning in air quality, climate change, and the composition of the troposphere. It is, however, the later paper "Biomass Burning in the Tropics: Impact on Atmospheric Chemistry and Biogeochemical Cycles" by Crutzen and Andreae (Crutzen and Andreae, 1990), one of the top 10 most cited Atmospheric Chemistry papers (Table 1), that has had the greatest impact on this research area, providing quantitative estimates of the amounts of biomass burning taking place around the world and the resulting emissions, recognizing the critical role of biomass burning emissions in the Tropics and from activities in developing countries that were not well documented. Hao and Liu (1994) made a further advance, looking at where and when biomass burning and thereby the related emissions occur. They developed an improved database of the amount of biomass burned owing to deforestation, shifting cultivation, savannah fires, fuel wood use, and clearing of agricultural residues, focused on tropical America, Africa and Asia during the late 1970s.

Simoneit et al. (1999) introduced the important concept that "the monosaccharide derivatives (e.g. levoglucosan) are proposed as specific indicators for cellulose in biomass burning emissions." They showed that levoglucosan is emitted at such high concentrations that it can be detected in air pollution filter samples at considerable distances from the original combustion source, allowing for source apportionment.

The 2001 paper "Emission of trace gases and aerosols from biomass burning" by Andreae and Merlet (2001), which is also one of the top 10 most cited papers (Table 1 and Figure 15), pulled together emission factors for a large variety of species emitted from biomass fires and is considered a key reference for biomass burning emission factors. Further work in the biomass burning area was later presented by Reid et al. (2005) in a review paper where they looked at measurements of smoke particle size, chemistry, thermodynamic properties, and emission factors from a variety of sources, including laboratory burns, in-situ experiments, remote sensing and modelling. They brought together information from the 'milieu of small pieces of the biomass-burning puzzle' and showed that there are large differences in measured particle properties and particle carbon budgets across the literature. van der Werf et al. (2006) investigated interannual variability and the underlying mechanisms regulating variability at continental to global scales using a time series of eight years of satellite and

model data. Total carbon emissions was driven by burning in forested areas, while the amount of
burned area was driven by savannah fires, which are influenced by different environmental and
human factors than forest fires.

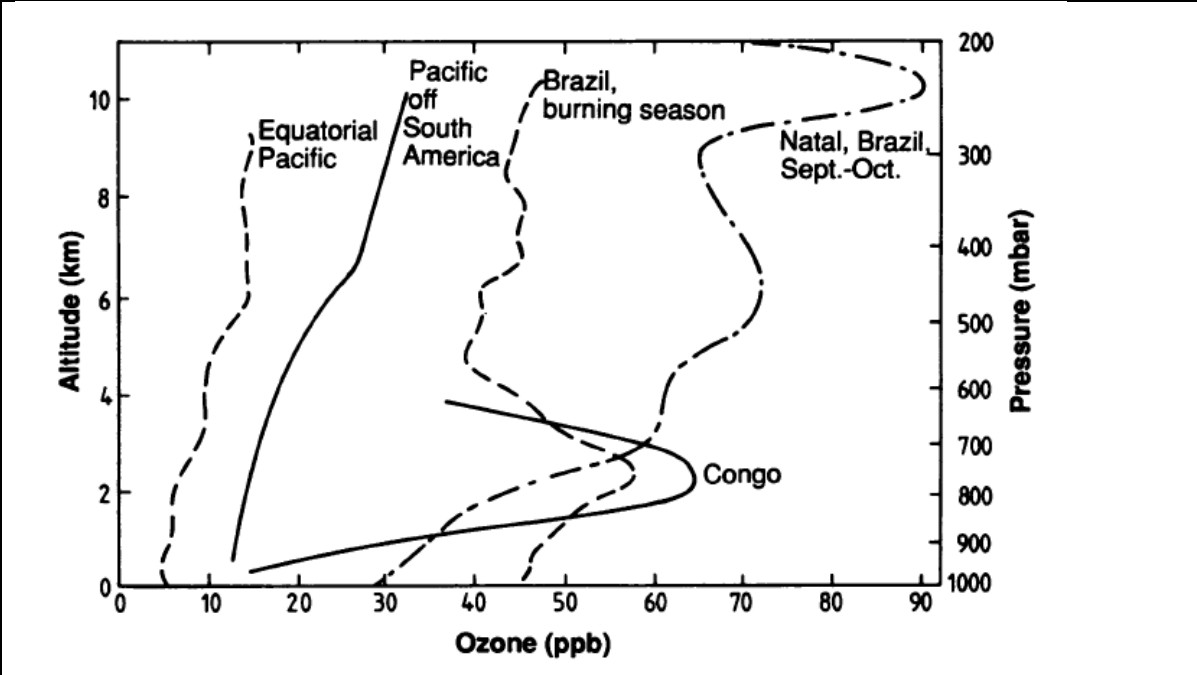

Figure 15 - Vertical profiles of O₃ in the tropical troposphere. The profile over the equatorial Pacific shows no influence from biomass burning, whereas the profile over the Pacific off South America suggests O₃ enhancement due to long-range transport from the tropical continents. The O₃ profiles over Brazil and the Congo show high O₃ concentrations at altitudes between 1 and 4 km due to photochemical production in biomass burning plumes. At higher altitudes, O₃ concentrations are also substantially enhanced, possibly also because of O₃ production by reactions in the emissions of biomass burning (Crutzen and Andreae, 1990).


*2.15.     Emissions and Deposition*
Non-chemical sources and sinks of various species are critical components of atmospheric processes
and therefore are particularly essential for global and regional models. An early advocate for such was
the work of Olivier and the team that created the EDGAR (Emission Database for Global Atmospheric
Research) database (Olivier et al., 1994).
Emissions from vehicles and power plants have always been an essential aspect of air quality related
policies, and therefore an area where more focused inventory work was needed and done. This
approach was pioneered in California in the early 1990s, and the studies by Calvert et al. (1993),
Lawson (1993) and Singer and Harley (1996) helped define and verify the California Smog Control
Program, providing a solid scientific basis with reliable emissions data. Techniques such as the remote
monitoring of traffic generated carbon monoxide (Chaney, 1983;Bishop and Stedman, 1996) are also
essential to recognise for the role they played in understanding vehicle emissions.
Agricultural emissions (from both crop and animal production) play an important role in several
atmospherically mediated processes of environmental and public health concerns (Chameides et al.,
1999). These atmospheric reactions and processes affect local and regional environmental quality,
including odour, particulate matter (PM) exposure, eutrophication, acidification, exposure to toxics,
climate, and pathogens (Erisman et al., 2008;Aneja et al., 2009). Agricultural emissions also contribute
to the global problems caused by greenhouse gas emissions, specifically nitrous oxide and methane.
The deposition of gases and aerosol particles to the surface is another critical process in the
atmosphere. Chamberlain (1966) is credited with the first exposition of the resistance network
approach to describe the uptake of gases on surfaces and the identification of transport through the
atmospheric boundary layer, through the surface layer, and through the stomata on plants, as
important elements of surface uptake. Building upon this work, a comprehensive and widely adopted,
parametrization of the dry deposition process for regional and global models was presented in the
late 1980s by Wesely (1989). Currently, there has been renewed interest in quantifying and
understanding deposition processes. Yet, a systematic description based on fundamental
independently measureable physico-chemical parameters is lacking. A complication can arise from a
range of oxygenated VOC that can exhibit bi-directional exchange above vegetation (Karl et al., 2010).

2.16.   *Chemical Transport*
Transport is an integral part of atmospheric processes and influences atmospheric composition across
a range of spatial scales. As early as 1975, Junge (1975) pointed out the importance of the atmospheric
residence time of a constituent with respect to global transport and dispersion. Prather's work
(Prather, 1994, 1996) provided new insights into timescales for atmospheric oxidation chemistry.
Stratospheric-tropospheric exchange (STE) has always been recognised as a key mechanism in
determining tropospheric composition. Early chemical dynamics were demonstrated by Danielsen
(1968) that laid the foundations for 3-dimensional modelling of chemical transport looking at
stratosphere-troposphere exchange based on radioactivity, ozone, and potential vorticity. Later,
Holton and co-authors (Holton et al., 1995) proposed an approach that placed stratosphere-
troposphere exchange in the framework of general circulation and helped clarify the roles of the
different mechanisms involved and the interplay between large and small scales, by the use of
dynamical tracers and potential vorticity. This work is recognised as a big step forward for the
understanding of the tropospheric ozone budget. Stohl and colleagues (Stohl et al., 2003) brought
together what has been viewed by many as the authoritative work on stratosphere-troposphere
exchange.
While the regional nature of air pollution has always been recognised, that is less the case for the
impact of trans-continental emissions on air quality. Jacob et al. (1999) showed that there was a need
for a global outlook for understanding regional air quality and meeting pollution reduction objectives.
This perspective spawned a decade of intense work on intercontinental air pollution and transport.
Well recognised in this area is the work by Stohl and colleagues (Stohl et al., 2002) who mapped out
the pathways and timescale of intercontinental air pollution transport and brought life to the subjects
of atmospheric dynamics and transport of air pollution (Figure 16).   Observational studies such as
Merrill et al. (1985), Moody et al. (1995), Stohl and Trickl (1999) and Forster et al. (2001) showed the
range of mechanisms and impact of long-range transport of air pollution.


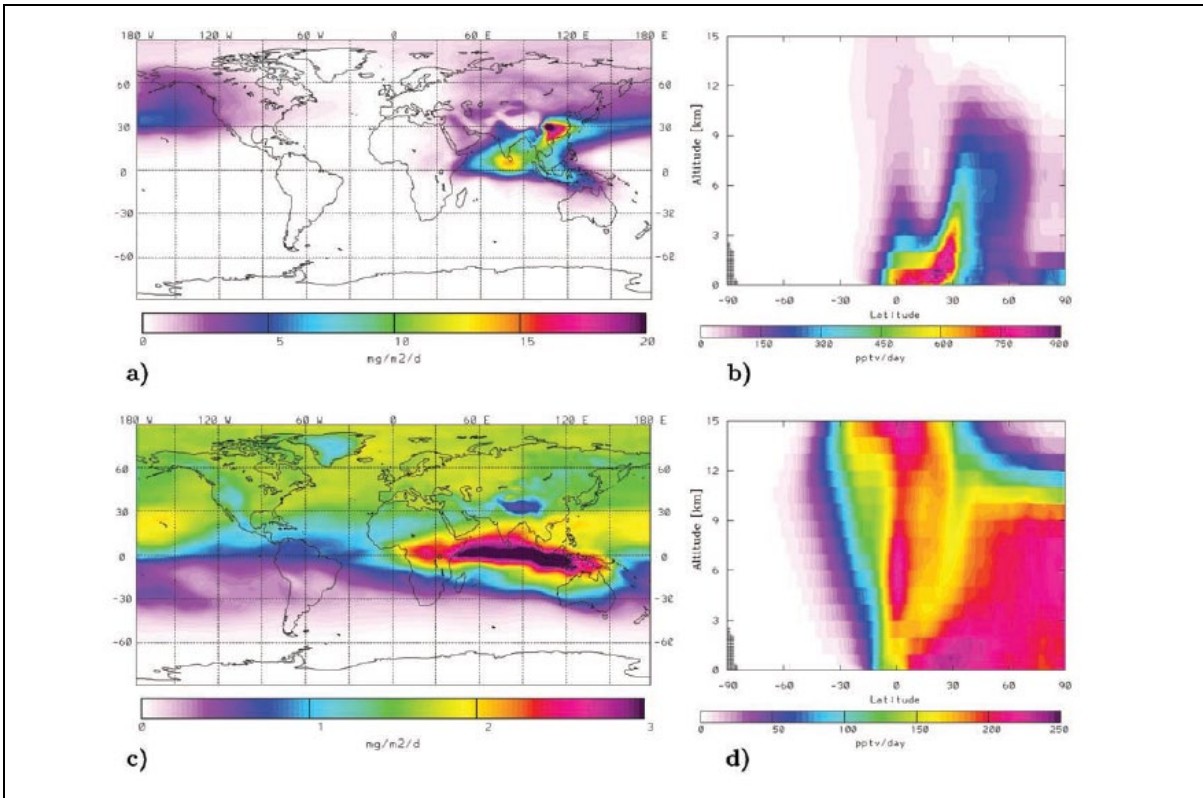

Figure 16 - Total columns (a, c) and zonally averaged mixing ratios (b, d), both divided by the respective time interval, of a Asia tracer for ages of 6 – 8 days (a, b) and 25– 30 days (c, d) during DJF.  The plots shows the horizontal and vertical impact of a pollution tracer (Stohl et al., 2002).


Moody et al. (1998) explored atmospheric transport history using back trajectories for the Harvard
Forest experiment demonstrating the power of trajectory methods at the regional scale. Key tools in
the development of this area were, the early simplistic isentropic trajectory methods (Merrill et al.,
1985),  the Hybrid Single-Particle Lagrangian Integrated Trajectory (HYSPLIT) model (Draxler and Hess,
1998), a forerunner of various particle trajectory and dispersion models, which developed into a
widely used particle dispersion model, and the FLEXible PARTicle dispersion (FLEXPART) model, a
Lagrangian particle dispersion model designed for calculating the long-range and mesoscale dispersion
of air pollutants (Stohl et al., 2005).
The ideas of intercontinental source-receptor relationships were embodied in the aforementioned
works by Jacob et al. (1999) and Stohl et al. (2002). The long-range transport concept was developed
in an effort to explore the source-transport relationships that drive observed ozone concentration in
regions farther away from the emission regions (Fiore et al., 2009) (see the 2.6. *Chemical Models*
section). This large community collaboration (Fiore et al., 2009) provided valuable insights into the
sensitivities of the hemispheric regional background of ozone and how this is controlled by emissions
from continental source regions.
Another critical area for atmospheric chemistry is boundary layer dynamics and meteorology. This is
particularly important since most emissions are emitted in the boundary layer. Atmospheric dynamics
in this important region have been mostly expressed as parameterizations in numerical models (Stull,
1988).  The spatial and temporal scales involved in the processes in this region cover a wide range.
The understanding of this region has been mostly based on meteorological and energy/water vapour
balance points of view. However, the chemical transformation and dispersion in this region are crucial
for how much chemicals actually get out of this region to influence the regional and global
atmosphere. Furthermore, the process of dry deposition, a critical loss processes for chemicals, is
mostly limited to the boundary layer.

*2.17.        Satellites and the Troposphere*
The importance of satellites for the discipline of atmospheric chemistry centres on the ability to give
a self-consistent global view of a selected set of tropospheric trace species (Burrows et al.,
2011;Martin, 2008;Prospero et al., 2002). The beginning and first demonstration of the effective
application of these attributes for the troposphere were the data and the retrievals from the GOME
instrument (Burrows et al., 1999) on ERS-2 and SCIAMACHY on Envisat (Bovensmann et al., 1999).
Historically, the roots of these early instruments are in stratospheric chemistry, with GOME being
deployed to be able to track stratospheric ozone and its key controlling chemical species.  Much effort
has  flowed with measurements from instruments such as OMI, MOPITT, TES, MODIS and ACE as well
as shuttle borne instrumentation, e.g. CRISTA (Burrows et al., 2011;Martin, 2008).
The most nominated paper in this area, and one that demonstrated the power of such observations
for tropospheric composition research, was "Increase in tropospheric nitrogen dioxide over China
observed from space" by Richter et al. (2005), which showed the capability of the satellites to track
the build-up of air pollution over vast regions from space (Figure 17). The importance of the work lead
by Palmer and co-workers in establishing a method to convert satellite observations to vertical
columns for comparison with e.g. models was also widely recognised (Palmer et al., 2001).

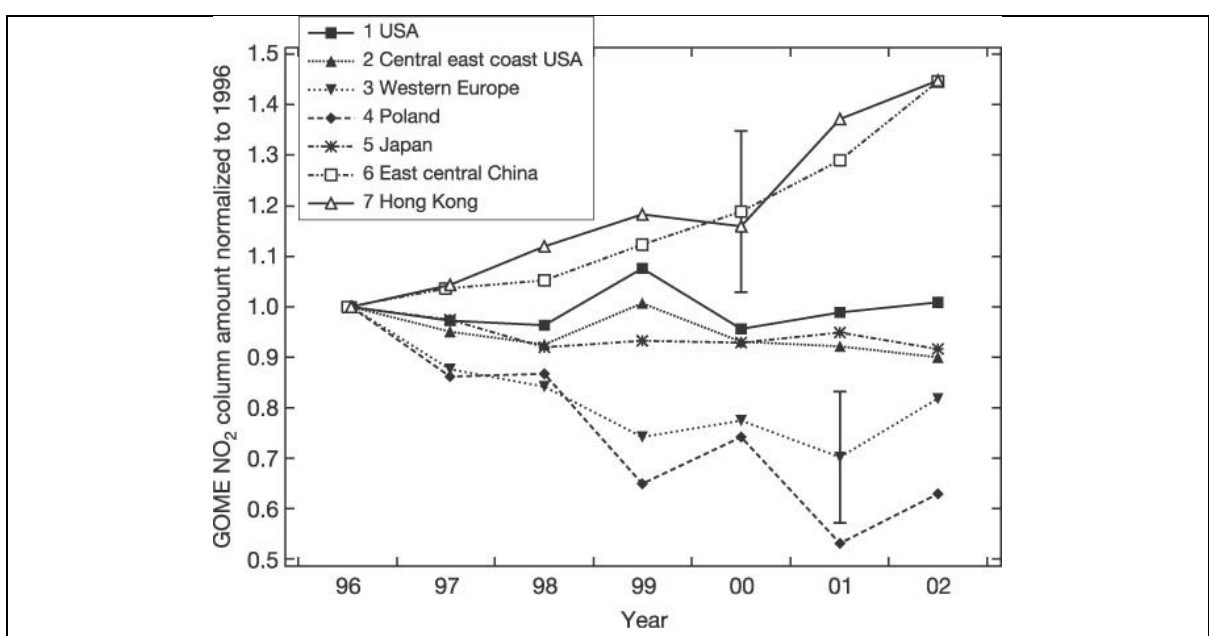

Figure 17 - The mean annual $NO_2$ column amount normalized to that in 1996 for the geographical
regions USA, Central East Coast USA, Western Europe, Poland, Japan, East Central China, and Hong
Kong showing a marked increase in $NO_2$ over China and decrease over Europe (Richter et al., 2005).


Satellite-based instrumentation can measure not only gas-phase trace species but also dust and
aerosol particles; mapping of the global distribution of dust (Prospero et al., 2002), the combination
of SEAWIFs and TOMS to track Asian dust events (Husar et al., 2001) and the development of the
MODIS aerosol algorithm (Remer et al., 2005) provided convincing demonstrations of this capability.
A step-change in this area was made with the paper Wang and Christopher (2003) "Intercomparison
between satellite-derived aerosol optical thickness and $PM_{2.5}$ mass: Implications for air quality
studies" which was the first description of the derivation of surface $PM_{2.5}$ from satellite aerosol optical
depth (AOD), built on by Liu et al. (2004) for the USA. Later, a global picture was developed by van
Donkelaar et al. (2006). These type of observations have been extensively used to estimate the global
impact of particulate matter (both $PM_{2.5}$ and $PM_{10}$) on health.

### 2.18. Stratospheric Chemistry

Tropospheric chemistry has always been influenced by the study of stratospheric chemistry. At the
same time, tropospheric chemistry has been pivotal in determining what surface emissions get to the
stratosphere. As mentioned earlier (see the 2.1. *Foundations* section), the basis of stratospheric ozone
chemistry was laid in the 1930s (Chapman, 1930) (for a full history see Brasseur (2019)), whereas
tropospheric chemistry followed a couple of decades later. Interest in stratospheric ozone chemistry
increased substantially following the works of Johnston and Crutzen on the role of nitrogen oxides in
the stratosphere (Crutzen, 1970;Johnston, 1971). The impact of supersonic transport (Johnston, 1971)
and of chlorofluorocarbons (CFCs) (Molina and Rowland, 1974) were important events for
stratospheric ozone chemistry. While Crutzen (1970) showed that the nitrogen oxides in the
stratosphere come mostly from the nitrous oxides from the ground, Johnson suggested that a fleet of
supersonic aircraft could release large amounts of nitrogen oxides into the lower stratosphere causing
substantial ozone loss (Johnston, 1971). The potential threat of supersonic transport highlighted the
importance of gas-phase catalysis in the atmosphere and, in particular, of the catalytic ozone
destruction by $HO_x$ and $NO_x$. These works opened the world's eyes to the potential for global
environmental change from human activities. Soon after, Lovelock (1974) identified CFCs in the
troposphere and showed that practically all the CFCs emitted to date were still in the atmosphere. The
significant contributions of Hampson (1964), Crutzen, and Johnston, and the recognition of chlorine-
catalysed ozone destruction by Stolarski and Cicerone (1974), paved the way for the seminal work of
Molina and Rowland (Molina and Rowland, 1974) linking chlorofluorocarbons to ozone layer
depletion. The recognition that bromine compounds can also destroy stratospheric ozone (McElroy et
al., 1986) further refined the story. The potential role of iodine in stratospheric ozone depletion has
also been raised (Solomon et al.1994), but it is still somewhat unsettled.
The ozone hole (Farman et al., 1985) was an unanticipated shock that awoke the world to the global
nature of ozone layer depletion. The origin of the ozone hole was understood in an historic set of
studies over a relatively short five-year period. First was the insightful and seminal work of Solomon
et al. (1986) that showed that chlorofluorocarbons and other ozone-depleting gases were the key
anthropogenic ingredient for the ozone hole. The confluence of cold temperatures that lead to the
formation of polar stratospheric clouds (PSCs) and the winter vortex formation over Antarctica
provided the opportunity for the massive ozone depletion that resulted in the ozone hole. This work
confirmed the suggestion of Farman (Farman et al., 1985) that the ozone hole was due to the
increasing abundances of CFCs. In particular, Solomon and co-workers (Solomon et al., 1986)
recognized that stable molecules such as $ClONO_2$ and HCl could react on solids (and indeed liquids).
Along the way, during this intense investigative period, the detection and quantification of the role of
ClO as a catalyst by Anderson et al. (1991), as well as De Zafra et al. (1988), was the "smoking gun"
that linked the CFCs with the ozone hole. The entire set of field measurements, from the ground,
aircraft, and balloons, solidified this linkage.
Less heralded, but equally important, were the laboratory studies that showed that chlorine nitrate
($ClONO_2$) and HCl did indeed react on PSCs (Hanson and Ravishankara, 1994;Tolbert et al., 1987;Leu,
1988;Molina, 1991;Molina et al., 1987) and determined the critical rate coefficients for the self-
reaction of ClO, the rate-limiting step in the unique catalytic cycles in Antarctica (Cox and Hayman,
1988;Sander et al., 1989;Trolier et al., 1990). Much was learned in later years by studying the Arctic
and from the continued observations over the Antarctic. It should be noted that the termolecular
reaction of ClO was suggested to be important for the chlorine chemistry by Molina and Molina
(Molina and Molina, 1987), and the history of this reaction goes back to Norrish's work at Cambridge
(Norrish and Neville, 1934). One of the lessons from this episode is that natural factors, in this case
the formation of a vortex and the occurrence of polar stratospheric clouds, can lead to unexpected
consequences when an anthropogenic ingredient (ozone-depleting chemicals) is added to the mix.
The numerical modelling of the stratosphere was an important ingredient for the success of mitigating
polar ozone loss, along with the theories of the ozone layer depletion and the ozone hole, the
laboratory studies of key processes, and the measurements in the atmosphere. Over the years, these
models have enabled a great deal of understanding of the coupling between chemistry and climate.
The development of stratospheric chemical transport models (Chipperfield and Pyle, 1988) was a
pivotal advancement that enabled quantitative understanding of the ozone layer depletion, including
the ozone hole (see the 2.6. *Chemical Models* section).
The weight of science led to the Vienna convention, the Montreal Protocol, and the Protocol's many
amendments and adjustments that are leading to the phasing out of the ozone-depleting gases. The
Montreal Protocol is the first international treaty on an environmental issue to be universally ratified
and is regarded as one of the most successful. That said, the ozone layer depletion story is not
complete. For example, the recognition that nitrous oxide is the remaining major ozone-depleting gas
emission (Ravishankara et al. 2009) has connected food production (tropospheric nitrogen cycles) to
ozone layer depletion and highlighted the importance of a holistic approach to environmental issues.

*2.19 Other issues that influence tropospheric chemistry*
Atmospheric chemistry advances have been influenced by growth in other areas. In particular, the
importance of anthropogenic climate change has been instrumental in invigorating atmospheric
chemistry studies because many of the major climate forcing agents are chemically active and climate
change, in turn, influences tropospheric chemistry (von Schneidemesser et al., 2015). In addition to
climate change, other adjacent discoveries and findings have influenced tropospheric chemistry
studies.
The global atmospheric and climatic consequences of nuclear war were investigated by both Crutzen
and Birks (1982) and Turco et al. (1983). Using models developed for looking at the impact of volcanic
eruptions, Turco et al. (1983) concluded that "enhancement of solar ultraviolet radiation due to ozone

depletion, long-term exposure to cold, dark, and radioactivity could pose a serious threat to human survivors and to other species." Similarly (Crutzen and Birks, 1982) concluded that "the screening of sunlight by the fire-produced aerosol over extended periods during the growing season would eliminate much of the food production in the Northern Hemisphere".

Air quality has an obvious direct impact on people, and this connection was recognized very early (it was in fact the primary motivation behind the fundamental work of Haagen-Smit). In 1993, Dockery et al. (1993) presented a study of six US cities showing a direct association between air pollution and mortality rates. This paper is a great example of how an adjacent field influences another, in this case atmospheric chemistry and public health. Though association between air pollution and health stretch back to the Los Angeles and London smog, the 'Six Cities Study' was a landmark as it demonstrated that the association between air pollution and mortality extended to much lower concentrations than those observed in the smog days.

### 3. Discussion and Summary

A mixture of the history of the discipline and its landmark ideas emerges from this exercise of asking the community what they consider to have shaped their research field. Table 2 seeks to bring these elements together to look at the evolution of the leading scientific concepts, their relevance to the environmental legislation (in this sense, we acknowledge an Euro-/US-centric bias), and the most notable environmental events that have shaped the discipline. Atmospheric science often sits at an interesting intersection between the societal interests (e.g. acid precipitation, air quality, ozone layer depletion, and climate) and its scientific venture. Monks and Williams (2020) have recently explored how environmental events in air quality drive policy and how a scientific and societal paradigm shift occurs once the emergency phase has passed.

From an overview of all the nominated papers, several general features are apparent. Ambient measurements are one of the cornerstones of atmospheric science (Abbatt et al., 2014). It is clear that the atmosphere is under-sampled, but over time we have found many ingenious ways to build different measurement strategies from the ground, ships, aircraft, balloon, sonde and satellites. With a focus on chemistry, it is clear that one needs to be able to measure with surety, sensitivity, specificity and speed in the troposphere. Many of the nominated papers reflect the importance of instrument development. Examples include the electrostatic sizers in aerosol science (Knutson and Whitby, 1975), various techniques to measure the hydroxyl radical (Eisele, 1994;Stevens et al., 1994;Perner et al., 1976a), chemiluminesence for $NO/NO_2$ (Kley and McFarland, 1980), the development of chemical ionization mass spectrometry (Lindinger et al., 1998), the application of the GCxGC-MS technique (Lewis et al., 2000), and aerosol mass-spectrometry measurements, e.g., Zhang et al. (2007). Often, the science underlying the development of these instruments, such as ion-molecule chemistry, is not necessarily acknowledged in the community. The paradigm for field instruments has been the development of analytical methods in the laboratory that are then adopted and/or adapted for field studies. The advances in associated fields such lasers, optics, optical detectors (e.g. camera and diode arrays), mass spectrometry (such as ion-traps and high-resolution time-of-flight mass spectrometry), separation methods (such a various chromatography methods), and meteorological instruments have fundamentally altered or understanding of atmospheric chemistry in general and tropospheric chemistry in particular. For example, the recent developments in chemical ionization mass

spectrometry have led to their pervasive use in our science. It is fair to say that the ability to separate and measure constituents in the part-per-quadrillion and part-per-trillion mixing ratio range have led to the detections of miniscule amounts of chemicals and their variations. Similarly, the recent revolution in low-cost sensors coupled with the evolution in telecommunications already is and will continue to change our field in the near future. Yet another important area is the development of the details of the chemistry through observations of intermediates and products, etc. These studies have provided some of critical information regarding the details of the chemistry (Cohen and Murphy, 2003).

Another common theme is the critical importance and impact of long-term observations, often termed monitoring, of key atmospheric components, from $CO_2$ (the "Keeling" curve) to chemically active molecules such halocarbons (the Antarctic ozone hole, ozone layer depletion and climate change), methane (the changes in the global OH field, background ozone production), and $NO_x$ (catalyst for tropospheric ozone production, vehicle emissions and acid precipitation). On the other hand, many breakthroughs in understanding the observations emerged because of basic laboratory information on kinetics and photochemistry (e.g., the reaction of $HO_2 + NO$, the determination of $O(^1D)$ quantum yields, and the reactions of $ClONO_2$ and $HCl$ on PSCs.) It is noteworthy that both laboratory studies and long-term observations are currently under funding stress, a situation that is already worrying the community (see, for example, the discussion in Burkholder et al. (2017)).

There is no doubt that atmospheric chemistry is an integrative science: one of the recurring themes in the papers discussed here is the tight relationship between ambient observations, laboratory experiments, and modelling. The integrative power of models has been recognised from the early studies by Levy (1971) to the development of highly sophisticated global transport models by Chipperfield and Pyle (1988) and Bey et al. (2001) up to the more recent demonstrations of the power of model ensembles (Stevenson et al., 2006;Fiore et al., 2009). Much of this progress has parallels in stratospheric chemistry. It is evident in the community that models are a powerful tool to map, test, and predict the atmosphere's past, present, and future. The predictions and projections from these models play essential roles in policy, planning, and management of environmental issues.

Another form of integrative or meta-analysis brings together a range of individual studies to produce a more significant outcome, such as new insights or models. There are some notable examples of this approach in the works on biomass burning by Crutzen and Andreae (1990) and more recently, by Van Der Werf et al. (2010). Similarly, the work to produce isoprene emissions models brought together several global isoprene flux measurements (Guenther et al., 1995;Guenther et al., 2006). Other examples include the work of Zhang et al. (2007) and Jimenez et al. (2009) who integrated various sets of AMS observations to give insight into land distributions of SOA.

There have been developments in fields adjacent to atmospheric chemistry that have shaped atmospheric chemistry progress. Examples include developments in epidemiology (e.g., the Six Cities Study by (Dockery et al., 1993)), atmospheric dynamics (the role of transport in determining chemical composition, the role of the Antarctic vortex), ocean science (pertaining to deposition to the ocean surface and emissions from the oceans), and in biological/plant science (e.g. (Kesselmeier and Staudt, 1999)). Integrating atmospheric chemistry with these adjacent fields is not only essential but also fruitful and was for many years embodied in the IGBP (Seitzinger et al., 2015), the World Climate Research Program (WCRP) and Earth-system science (now Future Earth) programs. Wider contexts, such as paleoclimate, have allowed an understanding of climate and atmospheric history over the

100,000 year timescales (e.g. (Petit et al., 1999)). That work set the framework for understanding that
the present-day atmospheric burdens of carbon dioxide and methane as important greenhouse gases
are unprecedented during the past 420,000 years. They also allow us to estimate the composition of
the troposphere in those ancient times.
Similarly, there are concepts that have their roots in tropospheric chemistry and have gone on to have
wider impact. The concept of the Anthropocene, most recently highlighted by Crutzen and Stroemer
in 2000, indicates that we are in a new geological epoch driven by human activities. The idea was more
fully expounded in Crutzen (2002) "the geology of mankind". There is little doubt that this has been a
key idea that has influenced much thinking as well as work far beyond atmospheric science (Table 2).
As discussed in detail in the introductory text, we opted here to assemble this compilation of papers
by reaching out to the community for nominations, rather than using the number of citations as a
primary measure. We will not go into the details again here, suffice it to say that there are inherent
advantages and drawbacks to any method one might consider for such a work. In that sense, we would
like to acknowledge and thank all those who provided feedback during the open peer-review phase
of the manuscript.
We would be remiss in not noting that there remain many areas of tropospheric chemistry that are
still in their infancy. The chemistry in the boundary layer and the dynamics of this region is one such.
This is particularly noteworthy since humans live and emit in the boundary layer. Another such
noteworthy sub-area is deposition which still largely consists of parameterizations. Developing an
understanding of the fundamental steps that are independently measured and understood would be
a laudable goal for atmospheric chemists. The state and development of emission inventories so key
to models remains an area in need of work. New measurements will provide insights into the ever
changing nature of our atmosphere across differing spatial and temporal scales.
The papers highlighted capture a substantial scope of the atmospheric research endeavour over the
last 60 years. The challenge now for you, the reader, is to continue to reflect on the papers included
here and continue this discussion for not only tropospheric chemistry but the related areas.

**4. Author Contributions**
PSM developed the concept and led the writing, PSM, ARR, RS, and EvS solicited input, ARR, RS, EvS
contributed to writing and editing.

Acknowledgments:

Work of ARR was supported by Colorado State University. The work of EvS was supported by the IASS
Potsdam, with financial support provided by the Federal Ministry of Education and Research of
Germany (BMBF) and the Ministry for Science, Research and Culture of the State of Brandenburg
(MWFK).

**Table 1 – Top 10 Cited Atmospheric Chemistry Papers (Atmospheric+Chemistry)**
(Scopus, 27/3/20)

| # | Paper | Title | Citations |
|---|-------|-------|-----------|
| 1 | Guenther et al. (1995) | A global model of natural volatile organic compound emissions | 2760 |
| 2 | Andreae and Merlet (2001) | Emission of trace gases and aerosols from biomass burning | 2350 |
| 3 | Guenther et al. (2006) | Estimates of global terrestrial isoprene emissions using MEGAN (Model of Emissions of Gases and Aerosols from Nature) | 2175 |
| 4 | Jimenez et al. (2009) | Evolution of organic aerosols in the atmosphere | 1909 |
| 5 | Atkinson (2000) | Atmospheric chemistry of VOCs and NOx | 1773 |
| 6 | Crutzen and Andreae (1990) | Biomass burning in the tropics: Impact on atmospheric chemistry and biogeochemical cycles | 1686 |
| 7 | Van Der Werf et al. (2010) | Global fire emissions and the contribution of deforestation, savanna, forest, agricultural, and peat fires (1997-2009) | 1578 |
| 8 | Atkinson and Arey (2003) | Atmospheric Degradation of Volatile Organic Compounds | 1502 |
| 9 | Grell et al. (2005) | Fully coupled "online" chemistry within the WRF model | 1436 |
| 10 | Lelieveld et al. (2015) | The contribution of outdoor air pollution sources to premature mortality on a global scale | 1425 |


(Web of Science, 27/3/20)

| # | Paper | Title | Citations |
|---|-------|-------|-----------|
| 1 | Ramanathan et al. (2001) | Atmosphere - Aerosols, climate, and the hydrological cycle | 2278 |
| 2 | Andreae and Merlet (2001) | Emission of trace gases and aerosols from biomass burning | 2168 |
| 3 | Hallquist et al. (2009) | The formation, properties and impact of secondary organic aerosol: current and emerging issues | 1988 |
| 4 | Jimenez et al. (2009) | Evolution of organic aerosols in the atmosphere | 1844 |
| 5 | Crutzen and Andreae (1990) | Biomass burning in the tropics: Impact on atmospheric chemistry and biogeochemical cycles | 1603 |
| 6 | Atkinson (2000) | Atmospheric chemistry of VOCs and NOx | 1596 |
| 7 | Atkinson et al. (1992) | Evaluated Kinetic and Photochemical Data for Atmospheric Chemistry: Supplement IV. IUPAC Subcommittee on Gas Kinetic Data Evaluation for Atmospheric Chemistry | 1488 |
| 8 | Grell et al. (2005) | Fully coupled "online" chemistry within the WRF model | 1332 |
| 9 | Lelieveld et al. (2015) | The contribution of outdoor air pollution sources to premature mortality on a global scale | 1292 |

| 10 | Bey et al. (2001) | Global modeling of tropospheric chemistry with assimilated meteorology: Model description and evaluation | 1212 |



**Table 2 – Science, Regulatory and Environmental Landmarks of the 20th and early 21st Centuries**

| Decade | Science Landmark | Regulatory Landmarks[1] | Environmental Events |
|---|---|---|---|
| 1930 | Chapman Cycles and Stratospheric Chemistry | | |
| 1940 | | | 1943 – LA Smog |
| 1950 | Air Pollution | 1956 - UK Clean Air Act | 1952 – Great Smog of London |
| 1960 | | 1963 - US Clean Air Act | |
| 1970 | Supersonic Transport Stratospheric Chemistry Tropospheric Chemistry Air Pollution and Clouds | 1978 – Ban of Lead in Petrol (USA) 1979 - CLRTAP (UNECE) | 1974 – Observations of acid rain |
| 1980 | Ozone Hole Chemistry Halogen Chemistry Biogenic Chemistry | 1987 - Montreal Protocol 1980 - $SO_2$ directive (EU) | 1985 – Observations of the ozone hole |
| 1990 | Air Pollution and Health Satellite Observations of the Troposphere Long-Range Transport of Air Pollutants | 1992 – Euro 1 Emission standard 1992 - Ozone Directive (EU) 1997 - Kyoto Protocol 1999 - Goteborg Protocol 1999 – Ban of lead in petrol (EU) | 1991 - Mt Pinatubo eruption |
| 2000 | SOA Concept of Anthropocene | 2001 - NEC Directive (EU) | |
| 2010 | Air Pollution and Climate | | 2015 - Dieselgate |
| 2020 | | | 2020 - COVID 19 |


[1] For more details on the UK/EU perspective see (Williams, 2004;Maynard and Williams, 2018) and
for the USA perspective see (Jacobson, 2002); see also (Monks and Williams, 2020).
**Table 3** - A global budget for NOx (Logan, 1983)

| | $10^{12}$ gm N yr$^{-1}$ |
|---|---|
| Sources | |
| Fossil fuel combustion | 21 (14-28) |
| Biomass Burning | 12 (4-24) |
| Lightning | 8 (8-20) |
| Microbial activity in soils | 8 (4-16) |
| Oxidation of ammonia | 1-10 |
| Photolytic or biological processes in the ocean | <1 |
| Input from the Stratosphere | ≈ 0.5 |
| | |
| Total | 25-99 |
| | |
| Sinks | |
| Precipitation | 12-42 |
| Dry Deposition | 12-22 |
| | |
| Total | 24-64 |


**Appendix 1 (Data runs to 2020 as illustrative)**

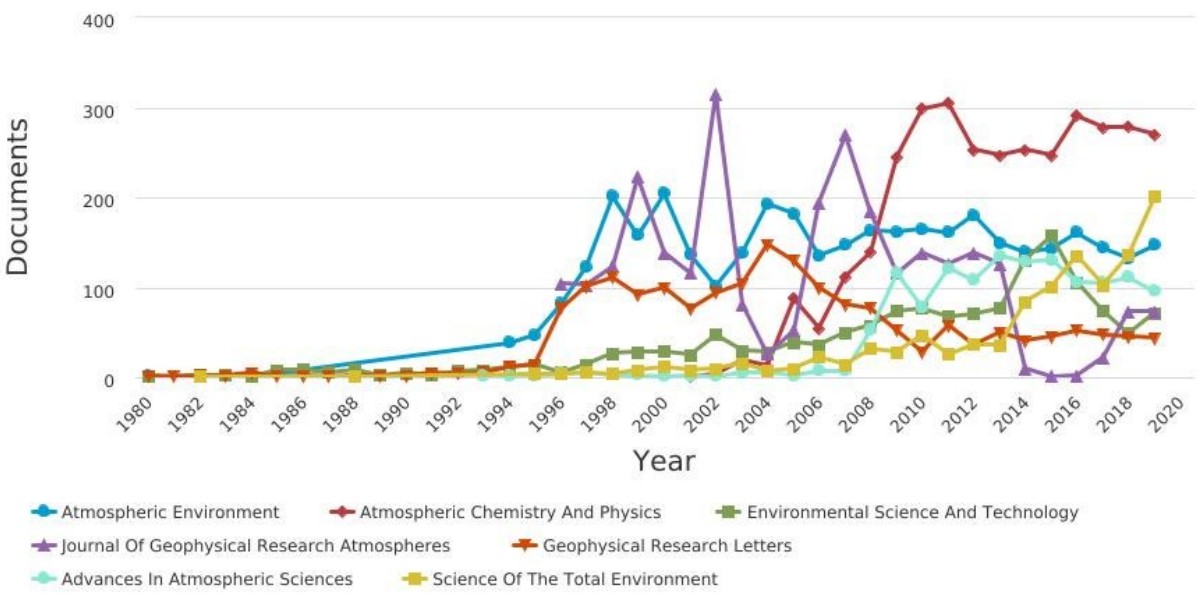



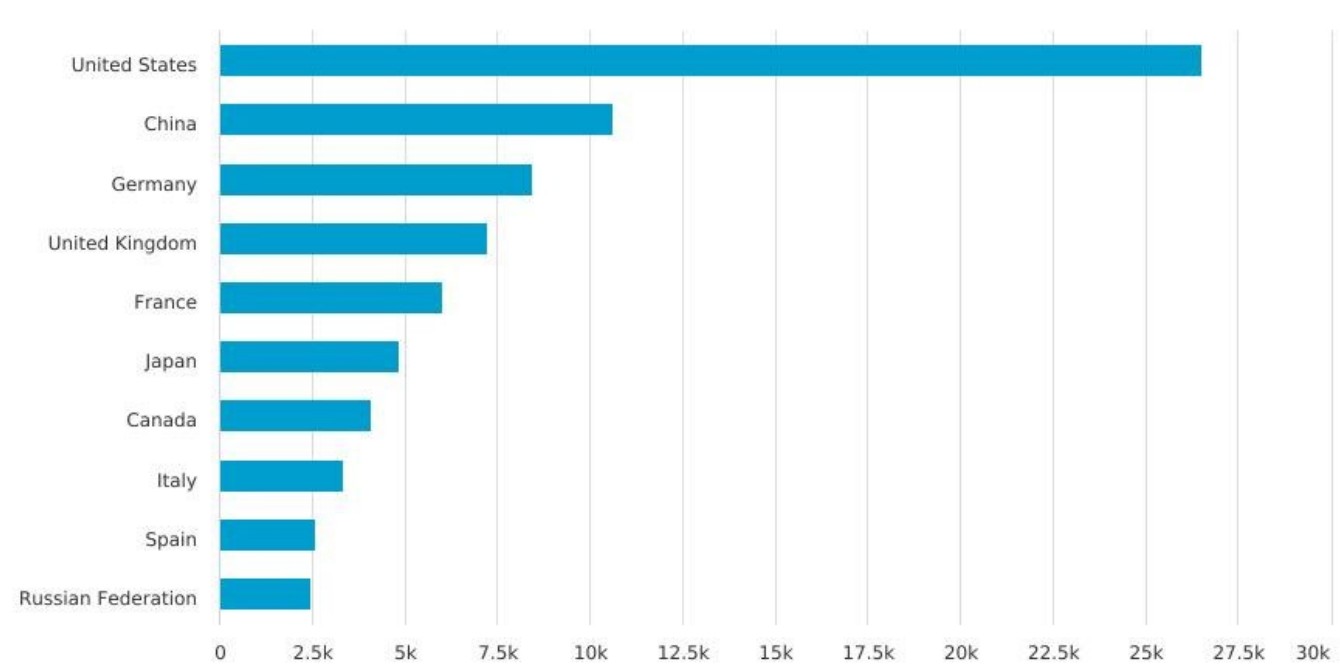


**Appendix 2 - Acronyms**

| | |
|---|---|
| ABLE | Amazon Boundary-Layer Experiment |
| ACE | Advanced Composition Explorer |
| AGAGE | Advanced Global Atmospheric Gases Experiment |
| AMS | Aerosol Mass Spectrometer |
| AOD | Aerosol Optical Depth |
| BBOA | Biomass-Burning Organic Aerosol |
| CAPRAM | Chemical Aqueous Phase Radical Mechanism |
| CCN | Cloud Condensation Nuclei |
| CFC | Chlorofluorocarbon |
| CLAW | Charlson-Lovelock-Andreae-Warren |
| CLRTAP | Convention on Long-range Transboundary Air Pollution |
| CRISTA | Cryogenic Infrared Spectrometers & Telescopes for the Atmosphere |
| CTM | Chemical Transport Model |
| EDGAR | Emission Database for Global Atmospheric Research |
| EPA | United States Environmental Protection Agency |
| ESRL | Earth System Research Laboratory |
| FA-AMS | Factor Analysis of Aerosol Mass Spectrometry |
| FLEXPART | FLEXible PARTicle dispersion model |
| GC | Gas Chromatography |
| GC-MS | Gas Chromatography-Mass Spectrometry |
| GEOS-Chem | Goddard Earth Observing System - Chemical Transport Model |
| GML | Global Monitoring Laboratory |
| GOME | Global Ozone Monitoring Experiment |
| HCFC | Hydrochlorofluorocarbon |
| HOA | Hydrocarbon-like Organic Aerosol |
| HYSPLIT | Hybrid Single-Particle Lagrangian Integrated Trajectory model |
| IAGOS | In-service Aircraft for a Global Observing System |
| IGAC | International Global Atmospheric Chemistry |
| IGBP | International Geosphere-Biosphere Programme |
| IPCC | Intergovernmental Panel on Climate Change |
| IUPAC | International Union of Pure and Applied Chemistry |
| JPL | NASA Jet Propulsion Laboratory |
| LIF | Laser Induced Fluorescence |
| LV-OOA | Low-volatility Oxygenated Organic Aerosol |
| MCM | Master Chemical Mechanism |
| MEGAN | Model of Emissions of Gases and Aerosols from Nature |
| MODIS | Moderate Resolution Imaging Spectroradiometer |
| MOPITT | Measurements Of Pollution In The Troposphere |

| MOZAIC | Measurements of OZone and water vapour by in-service Airbus airCraft |
|---|---|
| NASA | National Aeronautics and Space Administration |
| NEC | National Emissions reduction Commitments |
| NMHC | Non-Methane Hydrocarbon |
| NOAA | National Oceanic and Atmospheric Administration |
| OA | Organic Aerosol |
| OMI | Ozone Monitoring Instrument |
| OOA | Oxygenated Organic Aerosol |
| OPE | Ozone Production Efficiency |
| PAN | Peroxyacetyl Nitrate |
| PM | Particulate Matter |
| POA | Primary Organic Aerosol |
| PSC | Polar Stratospheric Clouds |
| RACM | Regional Atmospheric Chemistry Mechanism |
| RADM | Regional Acid Deposition Model |
| RF | Radiative Forcing |
| RRKM | Rice-Ramsperger-Kassel-Marcus |
| SAPRC | Statewide Air Pollution Research Center |
| SCIAMACHY | SCanning Imaging Absorption spectroMeter for Atmospheric CartograpHY |
| SeaWIFS | Sea-Viewing Wide Field-of-View Sensor |
| SHADOZ | Southern Hemisphere ADditional OZonesondes |
| SOA | Secondary Organic Aerosol |
| STE | Stratospheric-Tropospheric Exchange |
| SV-OOA | Semi-volatile Oxygenated Organic Aerosol |
| TES | Technology Experiment Satellite |
| TOMS | Total Ozone Mapping Spectrometer |
| UNECE | United Nations Economic Commission for Europe |
| UV | Ultraviolet |
| VOC | Volatile Organic Compound |
| WCRP | World Climate Research Program |
| WRF | Weather Research and Forecasting model |
| WRF-Chem | Weather Research and Forecasting model coupled to Chemistry |

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
