# Peer review of "Opinion: Papers that shaped Tropospheric Chemistry"

_Atmospheric Chemistry and Physics, 2020_

## Community Comment (CC3)

January 5, 2021.

I wish to compliment and thank the authors for a thoughtful and succinct analysis of 'State of Science' of Atmospheric Chemistry. The manuscript was a delight to read and captures most of the important developments in the field.

However, in my judgement the manuscript misses on discussing Agricultural Air Quality and its impact on the atmospheric environment. Agricultural emissions (from both crop and animal production) play an important role in several atmospherically mediated processes of environmental and public health concerns. These atmospheric reactions/processes affect local and regional environmental quality, including odor, particulate matter (PM) exposure, eutrophication, acidification, exposure to toxics, climate, and pathogens. Agricultural emissions also contribute to the global problems caused by greenhouse (nitrous oxide and methane) gas emissions. Agriculture is the largest contributor of both ammonia and nitrous oxide burden in the atmosphere. Moreover, a number of nitrogen-, sulphur- and carbon-containing compounds, including ammonia, nitrogen oxides, hydrogen sulphide, volatile organic compounds, and hazardous air pollutants are emitted through agricultural operations. Ammonia, in particular, plays a role in a host of environmental problems (e.g., air quality, odor, climate change, soil acidification, eutrophication, biodiversity), often through interactions (i.e. atmospheric chemistry) with other compounds in the atmosphere. The manuscript misses out on the opportunity to highlight the emerging role of ammonia and other agricultural emissions in the atmosphere.

I hope the authors will find these comments useful as they revise the manuscript.

Some References:

Aneja, V.P., J.P. Chauhan, and J.T. Walker, "Characterization of Ammonia Emissions from Swine Waste Storage and Treatment Lagoons," Journal of Geophysical Research-Atmospheres, Vol. 105, pp. 11535-11545 (2000).

Aneja, V. P. et al. (eds). Proceedings, Workshop on Agricultural Air Quality: State of the Science (North Carolina State University, Raleigh, North Carolina, (2006) http://ncsu.edu/airworkshop/.

Aneja, V.P., W.H. Schlesinger, and J.W. Erisman, "Effects of Agriculture upon the Atmospheric Environment of the United States: Research, Policy and Regulations", Environmental Science and Technology, vol. 43, pp. 4234-4240 (2009).

Aneja, V.P., W.H. Schlesinger, and J.W. Erisman, "Farming pollution", Nature Geoscience, vol. 1, pp. 409-411 (2008).

Baek, B. H., Aneja, V. P., & Tong, Q. Environ. Pollut. 129, 89–98 (2004).

Erisman, J. W. et al. Environ. Pollut. 150, 140–149 (2007).

Erisman, J., Sutton, M., Galloway, J. *et al.* How a century of ammonia synthesis changed the world. *Nature Geosci* **1,** 636–639 (2008). https://doi.org/10.1038/ngeo325

Galloway, J. N. et al. Science 320, 889–892 (2008).

Steinfeld, H. et al. Livestock's Long Shadow: Environmental Issues and Options (Food and Agriculture Organization of the United Nations, Rome, 2006); available at: http://www.virtualcentre. org/en/library/key_pub/longshad/A0701E00.pdf

Viney.

--

Viney P. Aneja | Professor
Department of Marine, Earth, and Atmospheric Sciences | North Carolina State University
Post Office Box 8208 | 5136 Jordan Hall | Raleigh, NC 27695-8208
P: 919-515-7808 | F: 919-513-8814 | Email: VINEY_ANEJA@NCSU.edu
http://www.go.ncsu.edu/airquality

---

## Community Comment (CC6)

**continues from the previous submission**

1. The item in lines 354-356 is misplaced, since bioaerosols have nothing to do with secondary organic aerosol. This item should be transferred in the section concerning aerosol and clouds. Also, I believe that this short sentence does not do justice to the importance of the issue. In fact, Airborne bacteria, fungal spores, pollen, and other bioparticles are essential for the reproduction and spread of organisms across various ecosystems, and they can cause or enhance human, animal, and plant diseases. Moreover, they can serve as nuclei for cloud droplets, ice crystals, and precipitation, thus influencing the hydrological cycle and climate (Fröhlich-Nowoisky et al., Atmospheric Research, 182, 346-376, 2016). These issues should be emphasized and in the above review paper pre-2010 appropriate references can easily be found.

2. Table 2 reports the most cited papers according to Scopus and WoS. Several papers are common to the two databases although in a different ranking. One can notice immediately that the most cited paper according to WoS does not even appear in the Scopus list. Unless the authors want to go into the reasons for this striking result (which I would not suggest), my suggestion would be to choose one of the two databases to be reported excluding the other.

3. Appendix 1 shows very interesting results on journals and regions of the references reported in the paper. I would suggest to include an Appendix 2 reported similar statistics concerning the contributions received by the authors from the call through the IGAC community. Knowing the number of contributions, their percentages by country, by age, gender, etc. would in my opinion help interpreting the outcomes of the paper.

4. Connected to the two above points, I would suggest to compile a table of the top ten papers suggested by the respondents to the author's call. It would be interesting to compare with the table 2 results.

---

## Community Comment (CC7)

**Monks et al.: Comment on Section 2.13, Biogenic Emissions and Chemistry**

This section omits any mention of the pioneering work of the Georgian Guivi Sanadze, who was the first to identify the emission of isoprene from plants (Sanadze, 1957). Unaware of Sanadze's work in the USSR, Rasmussen and Went independently discovered isoprene emissions shortly after (Rasmussen and Went, 1965). Sanadze was also the first to show that isoprene emission rates are temperature dependent (Sanadze and Kursanov, 1966). I think the first three paragraphs require revision to reflect Sanadze's contribution and the subsequent evolution of bVOC research and atmospheric chemistry.

I therefore suggest that **Section 2.13 Paragraphs 1, 2 and 3** now read:

The first report that plants emit volatile organic compounds into the atmosphere was made in 1957 by the Georgian scientist Guivi Sanadze (Sanadze, 1957). Unaware of Sanadze's work in the USSR, Rasmussen and Went independently discovered isoprene emissions in 1965 (Rasmussen and Went, 1965). Sanadze was also the first to show that isoprene emission rates are temperature dependent (Sanadze and Kursanov, 1966). However, the relevance of biogenic VOC for atmospheric chemical processes was not immediately recognized. Although Tingey at the USEPA did note the potential for isoprene to play a role in regional air quality in 1978, this was not formalised until the ground-breaking work of Chameides and colleagues in 1988 (Chameides et al., 1988) and MacKenzie and colleagues in 1991 (MacKenzie et al., 1991).

In 1992 the seminal review of Fehsenfeld et al. (1992) brought the importance of isoprene and a very wide range of other VOCs of biological origin to the attention of the atmospheric chemistry community, opening up an entirely new branch of atmospheric chemistry. Over time, this became pivotal for major policy formulations to abate ozone pollution.

Underpinning the atmospheric chemistry research that Fehsenfeld et al. promoted, plant physiologists began working on understanding the biological and environmental controls on biogenic VOC emission rates. This allowed the development of relatively simple functions to predict the emissions of biogenic VOCs which resulted in the first spatially and temporally resolved global model of global emissions (Guenther et al., 1993). This soon evolved into more sophisticated high resolution global models (Guenther et al., 2000;Guenther et al., 1995), allowing for the emissions of biogenic compounds to be included in atmospheric chemistry models across all scales. Eventually, this work took the form of the widely used MEGAN (Model of Emissions of Gases and Aerosols from Nature) model (Guenther et al., 2006) which is still used in modern Earth system models today (Table 2 and Figure 14).

New references

MacKenzie. A.R., R.M. Harrison, I. Colbeck and C.N. Hewitt, 1991. The role of biogenic hydrocarbons in the production of ozone in urban plumes in south-east England. Atmos. Environ., 25A: 351-359.

Rasmussen R.A. & Went F.W. (1965) Volatile organic material of plant origin in the atmosphere. Proceedings of the National Academy of Sciences of the United States of America 53, 220.

Sanadze G.A. (1957) The nature of gaseous substances emitted by leaves of *Robinia pseudoacacia*. Soobshch Akad Nauk Gruz SSR19, 83–86.

Sanadze G.A. & Kursanov A.L. (1966) On certain conditions of the evolution of the diene C5H8 from poplar leaves. Soviet Plant Physiology 13, 184–189.
* * *
Nick Hewitt, Lancaster University (14th January 2021)

---

## Author Comment (AC1)

We would like to thank **Reviewer #2** for the insightful kind comments and suggestions. Please find our replies.

**Main comments:**

This wonderful paper represents a very important contribution because it shows the evolution of an important research field until year 2010. It will provide to the new generations of students and scientists an excellent historical introduction that will allow them to realize how the concepts have evolved and how science progresses. It is particularly interesting to understand that progress in atmospheric chemistry resulted from the complementarity between laboratory experiments, field measurements, data analysis and modeling. I like the paper very much. I will highlight its strengths but I will also make some suggestions to make the paper somewhat more inclusive. I find the paper a bit unbalanced towards process studies and laboratory work (which should be highlighted as they are) at the expense of field work and modeling.

Strength of the Paper. The paper covers a lot of material and reports a lot of important results that shaped the field of atmospheric chemistry. A lot of bibliographical material is provided. Several graphs and tables are very interesting including, for example, the documents by sources and year and by country. The two tables with the top 10 cited papers are useful. The grouping of 19 topics presented in Section 1.2 is excellent. The discussion about how the papers were selected and the limitations to this approach is very good. It shows that the methodology does not provide an absolute standard and that work not included in this list can also be excellent. The paper is well written and easy to read. I am sure that people starting their career will like to read the document.

Suggestions. As I said above, I find the paper to be a bit unbalanced towards fundamental processes at the expense of more global and regional investigations. This could be easily addressed by adding some references to a few important observational and modeling accomplishments. Sometimes, big advances related, for example, to filed campaigns are not presented as a single paper, but are dispersed in several papers, and so none of them has a highest citation rate. For example, a major accomplishment before 2010 has been the organization of several airborne campaigns. Not only the ER-2 measurements of the ClO/O3 reactions have been key for the ozone hole question, but also the numerous airborne field campaigns in the 1980's and 1990's (many NASA field campaigns) that have provided insight on global tropospheric chemistry, and in particular on questions related to the oxidation capacity of the atmosphere. A lot has been learned from field campaigns (such as PEM, Trace P, ACE, and several others) and so perhaps, the authors could mention large projects that have provided fundamental understanding of the functioning of the atmosphere at the global and regional scales. Other topics that illustrate major advances: Field campaigns to assess NOx produced by lightning and the importance of thunderstorms. Production of methane by rice paddies, particularly in Asia. Monitoring of methane and carbon monoxide by international networks. Systematic observations of ozone and CO from commercial aircraft.

We agree with the reviewer about the importance of field campaigns. Much of the early multi-national fieldwork was captured in the IGAC paper from Melamed et al. (2015). We expanded the discussion in the paper at the relevant points to highlight the observational contribution of such campaigns.

Perhaps a bit more should be said about space observations. Much progress has been done before year 2010. The measurement of CO from the Space Shuttle gave a first global view of the distribution of this species and highlighted the role of biomass burning, particularly in the tropics. Beside GOME, and SCIAMACHY, one should perhaps refer to other important satellites and instruments such as OMI, MOPITT, MODIS, TES, etc. since they provided again a global view that was very useful to understand global budgets and the role of transport.

We agree with the reviewer on this point. The section on Satellite observations has been expanded and references to some overview works that chart the development of instruments and their applications have been added.

Finally, the Section on modeling is a bit weak. Models have been key for the interpretating observations, analyzing the role of chemical processes and projecting future changes in the atmospheric composition. The first relatively detailed 3-D model of the troposphere (although a bit intermediate) were developed in Mainz (MONGUNTIA), in Brussels (IMAGES) and at Livermore (GRANTOUR). They were followed by major efforts to develop really comprehensive chemical transport models (GeosChem at Harvard, MOZART at NCAR, Uni. Of Oslo model, and MESSY in Mainz, CHASER in Japan, etc.

We agree with the reviewer and have added a new expanded section dedicated to chemical models to the paper.

For stratospheric work, which is also mentioned in the paper, 2-D models such as the model of Garcia and Solomon have been a key tool to implement the fundamental concepts of the stratospheric residual circulation (and related meridional transport) resulting from planetary wave breaking presented by Holton and McIntyre, and to assess how the residual circulation could explain the meridional transport of ozone and other tracers in the middle atmosphere.

Several 3-D stratospheric models were also developed for the stratosphere. One of the first ones was developed at MIT by Cunnold, Aleya and Prinn and at GFDL by Mahlman. Some more advanced models include WACCM developed at NCAR

The Stratospheric section has been updated to include a broader perspective on this subject including a review of stratospheric models.

Nothing is said about the first attempts to develop inverse modeling techniques and chemical data assimilation methodologies, which were initiated in the late 1990s.

A sentence and a reference to inverse modelling has been added to the text.

I do not want to provide a "list of what is missing" since the goal of the paper is to highlight bthe most cited papers, but I am making some general points that perhaps would be welcome, and I am not asking to the authors to necessarily including all of them in the text. These are just constructive suggestions.

We thank the reviewer for the suggestions, they are gratefully received and acted upon.

**Some minor remarks on the text:**

Around line 109: How many responses were provided in response your solicitation of the scientific community?

We have received over 50 responses overall.

Line 320; Can you add a reference and perhaps an estimate of the uncertainty about the -1.1 Wm-2?

A reference to the IPCC report has been added.

Figure 4 looks dark to me and low quality (it may be my printer).

The quality of this (and other figures) has been improved.

Lines 364 to 367: Difficult to understand if you are not specialized (amorphous solid state). Beside the fact that it challenged the traditional views, what are the consequences of this discovery?

The text has been clarified.

Line 539: Second or third? I thought that methane was second (and close to ozone).

This refers to the identification rather than the order of importance.

Figure 10: the figure does not read well. Complicated to understand with such a short caption. Also, on the y-scale, it seems strange to express an integrated flux in ppb. I guess that it is integrated over time, but for how long.

The legend has been improved.

Line 742: I am not sure that research is trying to count the numbers of VOCs present in the atmosphere.

One could argue that the first stage of science is to count things (e.g. frogs in ecological science). The point further down is made as to the impact on the carbon budget.

Line 949: It is appropriate to cite Hampson, but a reference to his report should be added.

The reference has been added.

---

## Author Comment (AC2)

We would like to thank **Reviewer #1** for the insightful kind comments and suggestions. Please find our replies.

**Main comments:**

This is a wonderful paper that will provide many future generations of atmospheric scientists some perspective on how the field developed and its current status. The limitations and approach taken are clearly stated and their emphasis on integrating all aspects of the field, ambient measurements, lab studies and modeling, is terrific. This reviewer has just a few suggestions and minor typo's etc.

   1. Line 34: Suggest adding at least one other review from the health community re: human health impacts, e.g. by Landrigan et al in Lancet, 2018. The introduction says it is only treating papers 2010 and earlier but cites a 2015 Lelieveld paper for health effects so I assume later papers than 2010 are being cited for background?

The reference Landrigan et al, 2018 has been added and wider contextual papers post 2010 are in scope..

   2. Throughout, the word "aerosols" is used to denote "particles". Historically, the engineers who developed much of this area and the instrumentation have defined aerosols as particles and the gas in which they are suspended (e.g. see Hinds book). This would be a good place to exhibit consistency and use "aerosol particles" throughout, or just "particles" if that is what they mean.

The text has been changed as recommended.

   3. Line 360: As the authors indicate in the introduction, the fundamental work on which significant advances are made often ends up not being cited as the field develops. The laser ionization MS technique I think originated with Sinha, Rev. Sci. Instrum. (1984), followed by papers coauthored with Friedlander. It might be appropriate to include one of these references along with Prather and Murphy.

The reference Sinha et al. 1984 has been added.

   4. Line 367: I would add for clarity at the end of the sentence on this line "that assumed low viscosity, liquid-like particles where exchange with the gas phase is fast".

The text has been changed as recommended.

   16. Line 445: They state that the "distinction between heterogenous and multiphase chemistry is not always clear". This would be a good place to define these terms.

A reference (Ravishankara, 1997) that discusses this distinction has been added to the text.

   17. Lines 453-469: These two paragraphs mix multiphase and heterogenous chemistry in a somewhat random order. I think they are using "heterogeneous" to mean reactions at surfaces and "multiphase" to mean reactions involving the uptake of gases into (and reaction in) the liquid phase. It would be good to define the terms first and then separate out the examples they give into the two bins.

The sentence "Often "heterogeneous" is taken to mean reactions at surfaces and "multiphase" to mean reactions involving the uptake of gases into (and reaction in) the liquid phase." has been added to the text.

18. Line 455-459: This cites the work of Akimoto et al on the photoenhancement of HONO formation but it seems some discussion of the dark formation of HONO should come first. The Pitts et al, Int. J. Chem. Kin. (1984) might be the first discussion of this in chambers. Finlayson-Pitts et al, PCCP (2003) is a review of the area to that point.

References to the work of Pitts and Finlayson-Pitts have been added to the text.

19. Lines 571-573: It reads as if Sillman (1999) was the first to put together ozone isopleths but this goes back into the 1950's or 60's, probably Haagen-Smit and Fox (1954). It would be good to cite the origins here as it is so important to our understanding even today.

The sentence has been clarified to point out that Sillman drew on previous work.

20. Lines 685-692: Shroeder and Urone (EST, 1974) were probably the first to suggest sea salt reactions as a photochemical source of chlorine atoms. It could be cited here and on line 704. A review by Cicerone in Rev. Geophys. Space Phys. (1981) would be a good review too.

Both references have been added to the text.

21. Line 846: It seems odd to state that emission inventories were pioneered in the 1990's. Surely they were being developed long before that, at least in California?

While it is true that emission inventories were developed earlier, we note that their growth and widespread application took place in the 90s.

22. Lines 925-927: Satellites have been really useful for tracking dust events but it seems that either here or in another section a reference to the seminal studies on dust transport of Prospero et al in the 1970's should be included.

Reference to Prospero's work in this area has been added to the Satellites section.

23. Line 988: "by now phased out the ozone-depleting gases". I think this is overly optimistic to say this, given recent measurements showing there seem to be emissions from Asia that were not expected. Maybe "are now phasing out ozone-depleting gases"...

The sentence has been changed to "are leading to the phasing out of ozone-depleting gases".

24. Table 1 seems quite Euro-centric, for example not including some of the big smog events in the U.S. and the passage of the Clean Air Act etc.

US smog events are included in the table, and the Clean Air Act is mentioned. While we recognize that the Table is Euro/US-centric some international landmarks are mentioned (e.g. Mount Pinatubo, Kyoto and Montreal protocols, etc…)

25. Table 2 is in two parts. It was not clear to this reviewer how they differ from each other in what they are supposed to illustrate.

The two parts of the table (it is actually Table 2) are derived from two different databases (Web of Science and Scopus), reflecting slightly different indexing methods. This is indicated in a new caption.

26. Table 3 seems out of context. It is very detailed on NOx but similar detail is not given for other species. I suggest omitting this.

We think Table 3 is an important finding of one of the landmark papers on the subject of nitrogen chemistry, outlining the complete nitrogen budget for the first time, so we would like to keep it in the paper.

27. Appendix 1 data go to 2020. Since the focus of the article is 2010 and prior years, some comment in the captions might be appropriate.

The 2010-2020 data are included to provide context. This has been clarified in the caption.

**Minor typos, grammar etc:**

1. Line 34: impacts.

2. Line 35: "latterly links to their climate" is very awkward. Maybe replace "latterly links to their climate" with "and climate"

3. 3.Line 178: "pollutions"

8. Line 626: "any" should be "many"?

9. Line 824: "..most cited paper-"

10. Line 885: Should "The study" be omitted so the sentence starts: "Moody et al (1998) ..."

11. Line 900: Spelling error "emmited"

12. The formatting of the references (no space between etc) makes them very hard to read but I suppose that will get straightened out in the final document.

The text has been amended as indicated.